# Tuning the selectivity of catalytic nitriles hydrogenation by structure regulation in atomically dispersed Pd catalysts

Zhibo Liu[1,2,9], Fei Huang [2,3,9], Mi Peng[4,9], Yunlei Chen[5,6,9], Xiangbin Cai [7], Linlin Wang[1,2], Zenan Hu[1], Xiaodong Wen [5,6], Ning Wang [7], Dequan Xiao [8], Hong Jiang [4], Hongbin Sun [1✉], Hongyang Liu [2,3✉] & Ding Ma [4✉]

The product selectivity in catalytic hydrogenation of nitriles is strongly correlated with the structure of the catalyst. In this work, two types of atomically dispersed Pd species stabilized on the defect-rich nanodiamond-graphene (ND@G) hybrid support: single Pd atoms (Pd$_1$/ND@G) and fully exposed Pd clusters with average three Pd atoms (Pd$_n$/ND@G), were fabricated. The two catalysts show distinct difference in the catalytic transfer hydrogenation of nitriles. The Pd$_1$/ND@G catalyst preferentially generates secondary amines (Turnover frequency (TOF@333 K 709 h$^{-1}$, selectivity >98%), while the Pd$_n$/ND@G catalyst exhibits high selectivity towards primary amines (TOF@313 K 543 h$^{-1}$, selectivity >98%) under mild reaction conditions. Detailed characterizations and density functional theory (DFT) calculations show that the structure of atomically dispersed Pd catalysts governs the dissociative adsorption pattern of H$_2$ and also the hydrogenation pathway of the benzylideneimine (BI) intermediate, resulting in different product selectivity over Pd$_1$/ND@G and Pd$_n$/ND@G, respectively. The structure-performance relationship established over atomically dispersed Pd catalysts provides valuable insights for designing catalysts with tunable selectivity.

---

[1] Department of Chemistry, Northeastern University, Shenyang 110819, P. R. China. [2] Shenyang National Laboratory for Materials Science, Institute of Metal Research, Chinese Academy of Sciences, Shenyang 110016, P. R. China. [3] School of Materials Science and Engineering, University of Science and Technology of China, Shenyang 110016, P. R. China. [4] Beijing National Laboratory for Molecular Sciences, College of Chemistry and Molecular Engineering and College of Engineering, and BIC-ESAT, Peking University, Beijing 100871, P. R. China. [5] State Key Laboratory of Coal Conversion, Institute Coal Chemistry, Chinese Academy of Sciences, Taiyuan 030001, P. R. China. [6] University of Chinese Academy of Science, No. 19A Yuanquan Road, Beijing 100049, P. R. China. [7] Department of Physics and Center for Quantum Materials, Hong Kong University of Science and Technology, Clear Water Bay, Kowloon, Hong Kong SAR, P. R. China. [8] Center for Integrative Materials Discovery, Department of Chemistry and Chemical Engineering, University of New Haven, 300 Boston Post Road, West Haven, CT 06516, USA. [9]These authors contributed equally: Zhibo Liu, Fei Huang, Mi Peng, Yunlei Chen. ✉email: sunhb@mail.neu.edu.cn; liuhy@imr.ac.cn; dma@pku.edu.cn

Amines, both primary and secondary amines, are important raw materials in the synthesis of bioactive molecules in pharmaceuticals synthesis. There are many methods developed and utilized in both industry and academia to obtain amines, such as amination of aryl halides or alcohols, reductive amination of aldehydes or ketones, hydroamination of olefins or alkynes, hydrogenation of nitriles, alkylative amination, and base-promoted mono-N-alkylation[1–5]. Among them, hydrogenation of nitriles has attracted extensive attention. However, owing to the high thermodynamic stability of nitriles, hydrogenation of nitriles to target chemicals is difficult. Moreover, catalytic hydrogenation of nitriles yields complex products, including primary, secondary amines, imines, tertiary amines, and even by-products from hydrogenolysis[6–8]. Therefore, it is quite desirable to develop highly effective catalysts, especially those that can precisely control the product selectivity towards nitrile hydrogenation. In this regard, metal-based homogeneous catalysts such as Ir[9], Rh[10], Ru[11], and Re[12] have been widely applied in catalytic hydrogenation of nitriles because of their good activity and selectivity. However, separating the homogeneous catalysts from the product mixture is difficult, which makes the recycle of these expensive catalysts less feasible. To circumvent the reusability problems, using heterogeneous solid catalysts is a possible solution[13–17]. Nevertheless, unlike homogeneous catalysts that possess controllable active sites and coordination environments, rationally tuning the products distribution over heterogeneous catalysts has always been a challenge, owing to their heterogeneity in morphology and active site distribution and the variation in local coordination environment. Therefore, developing well-defined heterogeneous catalysts with high activity and good selectivity is an essential way to explore the structure–performance relationship of catalyst in the hydrogenation of nitriles.

In recent years, fully exposed cluster catalyst (FECC) contains atomically dispersed metal atoms on the support as the catalytic active sites have received increasing attention. FECCs offer diverse surface sites formed by an ensemble of metal atoms, comparing with single-atom catalyst[18–22], it not only provides maximized atom utilization but also possesses rich active sites and easily identified coordination structures. FECC is so highly dispersed that all the metal atoms within it are available for the adsorption and transformation of reactants. As its stable metal loading can be higher than that of SAC, the FECC usually exhibits higher mass specific activity than the SAC, which is critically important for industrial applications[23–30], providing a feasible approach to study the relationship of structure–performance of the hydrogenation of nitriles.

In this paper, we fabricate two kinds of atomically dispersed catalysts, single Pd atoms ($Pd_1$/ND@G) and fully exposed Pd clusters with an average atomicity of three Pd atoms ($Pd_n$/ND@G), both of which are immobilized on the defect-rich nanodiamond–graphene hybrid support (ND@G)[31–33]. The catalytic performance in the transfer hydrogenation of nitriles with $NH_3 \cdot BH_3$ (AB) as the hydrogen donor was evaluated over $Pd_1$/ND@G and $Pd_n$/ND@G[16,34,35]. We found that secondary amines are preferentially obtained over $Pd_1$/ND@G, whereas primary amines are selectively generated over $Pd_n$/ND@G. To explain the differences, we further carried out density functional theory (DFT) calculations to elucidate the different mechanisms of the two Pd catalysts induced by structural variation and to better understand their drastically different catalytic performance.

## Results

### Synthesis and characterization of $Pd_1$/ND@G and $Pd_n$/ND@G.
The atomically dispersed $Pd_1$/ND@G and $Pd_n$/ND@G were prepared by a deposition-precipitation strategy. Except for different loading amounts of Pd species, no difference was observed in physicochemical structures of two samples (see Supplementary Table 1). The X-ray diffraction (XRD) patterns of $Pd_1$/ND@G and $Pd_n$/ND@G exhibit characteristic peaks of ND@G; no diffractions associated with Pd were observed, indicating that the Pd species were highly dispersed in these two samples (Supplementary Fig. 2a). The aberration-corrected high-angle annular dark-field scanning transmission electron microscopy (AC-HAADF-STEM) images display detailed structures of as-prepared Pd species on ND@G (see Supporting Information for the ND@G support, Supplementary Fig. 1)[36]. For $Pd_1$/ND@G (Fig. 1a, b), all Pd single atoms were uniformly distributed on the support. Due to some uncontrollable factors in the preparation process, it is inevitable that there are a few close Pd metal atoms in the $Pd_1$/ND@G catalysts. While in $Pd_n$/ND@G (Fig. 1c, d), the Pd species mainly exists as fully exposed Pd clusters as highlighted in Supplementary Fig. 2b. Clearly, the Pd species in these two catalysts are both atomically dispersed on the ND@G support[24,37,38].

X-ray absorption near-edge structure (XANES) and extended X-ray absorption fine structure (EXAFS) measurements were performed to study the electronic structure and coordination environment of Pd species in $Pd_1$/ND@G and $Pd_n$/ND@G catalysts. From XANES results, the valence states of $Pd_1$/ND@G and $Pd_n$/ND@G were thoroughly investigated. Notably, the valence state of $Pd_n$/ND@G and $Pd_1$/ND@G are between those of PdO and Pd foil (Fig. 1e), indicating that fully exposed Pd clusters and Pd single atoms are both slightly positively charged. Meanwhile, the Pd species in $Pd_1$/ND@G are more positive charged than that in $Pd_n$/ND@G. X-ray photoemission spectroscopy (XPS) measurements further confirmed the chemical states of the atomically dispersed Pd catalysts (Supplementary Fig. 3). The binding energy of Pd $3d_{5/2}$ in $Pd_1$/ND@G and $Pd_n$/ND@G are 335.5 and 335.0 eV, respectively. The shifts of binding energy suggest that the valence state of Pd species in $Pd_1$/ND@G is more positive than that in $Pd_n$/ND@G, agreeing well with the XANES observations, implying a stronger charge transfer between $Pd_1$ species and the ND@G support.

The EXAFS spectra (Fig. 1f) of $Pd_1$/ND@G only featured a major peak near 1.5 Å from the first coordination shell of Pd associated with Pd-C/O scattering, indicating the formation of isolated Pd atom in $Pd_1$/ND@G. For $Pd_n$/ND@G, an additional peak at about 2.4 Å from first shell Pd–Pd coordination emerges, indicating the formation of Pd clusters[37,38]. The trend can also be clearly resolved from the WT of Pd k-edge EXAFS oscillation results (Supplementary Fig. 4)[39]. From the fitting structural parameters (see Supplementary Table 2 and Supplementary Fig. 5), it is obvious that the Pd–C/O coordination number in $Pd_1$/ND@G was about 2.6, indicating each Pd atom is bonded to three C/O atoms. For $Pd_n$/ND@G catalyst, the Pd–C/O coordination number was 2.5, while the Pd–Pd coordination number was 1.9, suggesting that the fully exposed Pd cluster was composed by about three Pd atoms on average, agreeing well with the AC-HAADF-STEM images as displayed in Fig. 1 and Supplementary Fig. 2b. From those results, we can conclude that these two catalysts exhibit distinct difference in coordination environment as well as in electronic structure, which will surely affect their reaction behavior in catalytic reaction.

### Benzonitrile transfer hydrogenation reaction.
Subsequently, the catalytic transfer hydrogenation of benzonitrile (BN) was evaluated over the atomically dispersed $Pd_1$/ND@G and $Pd_n$/ND@G catalysts. As shown in Fig. 2a, several products are obtained in the transfer hydrogenation of BN. For the control experiment with pure ND@Gas catalyst, no hydrogenation reaction happened,

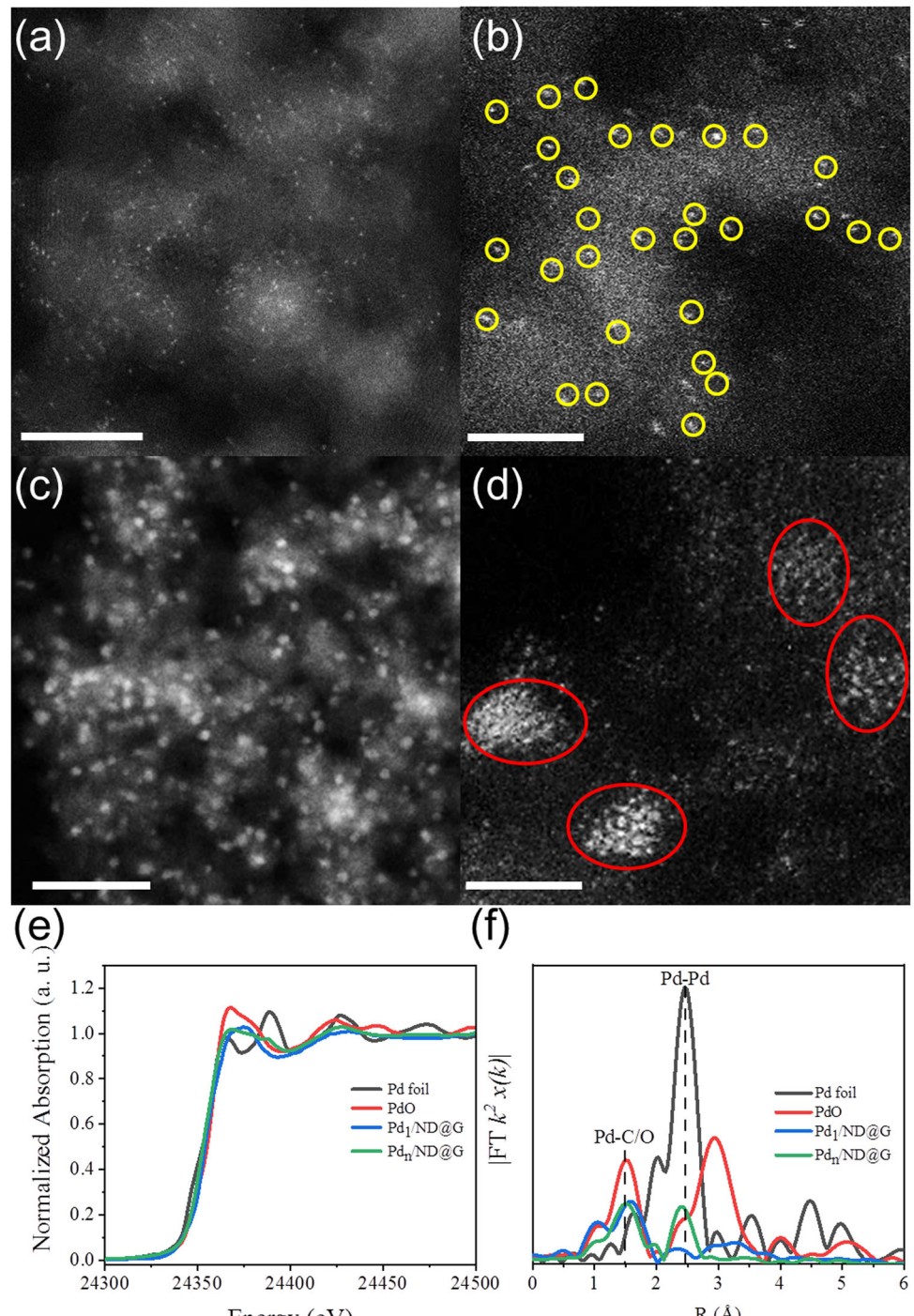

**Fig. 1 Structural characterization of catalysts. a** HAADF-STEM images of Pd$_1$/ND@G at low magnification. **b** Atomically dispersed single Pd atoms in Pd$_1$/ND@G highlighted by the yellow circles. **c** HAADF-STEM images of Pd$_n$/ND@G at low magnification. **d** Fully exposed Pd clusters in Pd$_n$/ND@G highlighted by the red circles. **e** Pd K-edge XANES profiles and **f** EXAFS spectra for Pd$_1$/ND@G and Pd$_n$/ND@G. Scale bars: **a**, 5 nm; **b**, **d**, 2 nm; **c**, 20 nm.

indicating that Pd is the active species instead of ND@G (Fig. 2e). Notably, Pd$_1$/ND@G and Pd$_n$/ND@G catalysts exhibited different catalytic performances (Fig. 2b, c). For Pd$_1$/ND@G catalyst under the optimized condition, the BN conversion is close to 100% after 8 h, producing dibenzylamine (DBA) with 98% selectivity (see Supplementary Table 3, entry 3 and Supplementary Table 4, entry 2). However, the DBA yield decreased slightly with extended reaction time, demonstrating that a small amount of DBA is converted to benzylamine (BA) (Fig. 2b). For Pd$_n$/ND@G, BN could be completely consumed within 0.5 h (Fig. 2c)

(see Supplementary Tables 5 and 6 entry 2), and the BA was primarily obtained (yield to BA above 98%, Fig. 2e). Further increase in the reaction time to 120 min, no transformation of BA can be observed, suggesting that it is the most stable product at this reaction condition over the fully exposed Pd cluster catalyst. And the commercial Pd/C catalyst was used as a reference (for the detailed structure information see Supplementary Fig. 6), although Pd/C catalyst also exhibited higher selectivity of BA, the catalyst activity was much worse compared with the fully exposed cluster Pd$_n$/ND@G catalyst (Fig. 2e).

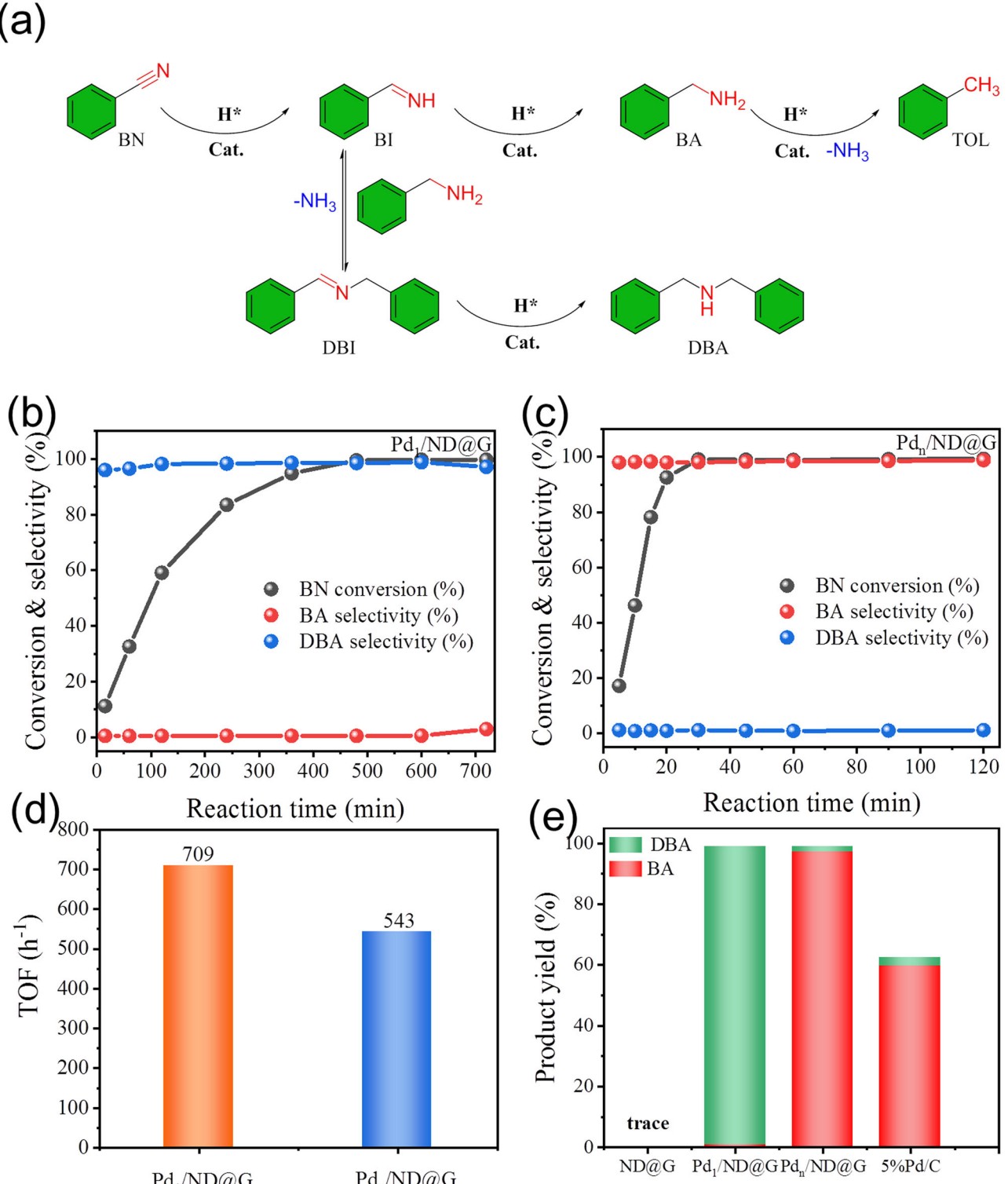

**Fig. 2 Catalytic performances of Pd1/ND@G and Pdn/ND@G catalysts in transfer hydrogenation of benzonitrile. a** Possible reaction scheme in the transfer hydrogenation of benzonitrile yielding benzylamine, *N*-benzylidenebenzylamine, and dibenzylamine. **b** Time-conversion plot for production formation from the transfer hydrogenation of benzonitrile over Pd1/ND@G. Reaction conditions: solvent, methanol, 10 mL; BN, 0.5 mmol; catalyst, 30 mg; AB, 4 mmol; temperature, 60 °C. **c** Time-conversion plot for production formation from the transfer hydrogenation of benzonitrile over Pd$_n$/ND@G. Reaction conditions: solvent, methanol, 10 mL; BN, 0.5 mmol; catalyst, 10 mg; AB, 3 mmol; temperature, 40 °C. **d** TOF over Pd1/ND@G (time, 15 min) and Pd$_n$/ND@G (time, 5 min). **e** Product yield for transfer hydrogenation of benzonitrile over ND@G, Pd1/ND@G (time, 8 h), Pd$_n$/ND@G (time, 30 min), and Pd/C (time, 30 min).

The catalytic performance results show that the transfer hydrogenation of BN is drastically different over the atomically dispersed Pd$_1$/ND@G and Pd$_n$/ND@G catalysts. BN shows high selectivity to secondary amines DBA over the single Pd atoms, while the fully exposed Pd clusters can selectively convert the BN to primary amines BA. As shown in Fig. 2d, the TOF@333 K of Pd$_1$/ND@G and the TOF@313 K of Pd$_n$/ND@G is 709 and 543 h$^{-1}$, respectively. The performance evaluation of the catalyst after the reaction was further evaluated; the activity of the Pd$_1$/ND@G catalyst was partially reduced. However, the Pd$_1$/ND@G catalyst still retains good selectivity of DBA (Supplementary Fig. 7e) and suggests that the Pd single atoms still play a major role. As shown in Supplementary Fig. 7f, there was no significant change in activity of Pd$_n$/ND@G catalyst after reaction, but the selectivity of DBA increased slightly. AC-HAADF-STEM measurements of the two catalysts after reaction further confirmed the leaching of a few Pd atoms leads to the decrease of the density of metal Pd in the Pd$_1$/ND@G catalyst after reaction (Supplementary Fig. 7a, b). And performance change of Pd$_n$/ND@G catalyst may be related to the decrease in the number of clusters and increase in the number of single atoms caused by the leaching of Pd in the Pd$_n$/ND@G catalyst (Supplementary Fig. 7c, d). Compared with previously reported catalytic systems, the as-prepared atomically dispersed Pd catalysts both show robust catalytic performance to corresponding amines under mild reaction conditions (see Supplementary Table 7).

**Substrate extension.** The reaction has been expanded to broad scopes of nitriles over the Pd$_1$/ND@G and Pd$_n$/ND@G catalysts, respectively. As shown in Fig. 3, Pd$_1$/ND@G and Pd$_n$/ND@G both show superior catalytic performance. As investigated, benzonitriles with electron-donating substituents and electron-withdrawing substrates obtain corresponding amines selectively. Compared with the aromatic nitriles, it was a great challenge to catalytically hydrogenate aliphatic nitriles, because it was less active and the by-products with methyl compounds are easy to produce in the hydrogenation[13,40]. Herein, phenylacetonitrile was taken as representatives and their catalytic hydrogenation over the Pd$_1$/ND@G and Pd$_n$/ND@G catalysts were discussed, respectively. The corresponding primary amines or second amines were acquired and the results were shown in Fig. 3. The results elucidate that the structure sensitivity over these two types of atomically dispersed Pd catalyst in hydrogenation are well kept in general nitriles. In short, our selectivity regulation strategy can be applied to a wide substrate range, and the two catalysts demonstrated a good tolerance to many functional groups.

**DFT calculations.** DFT calculations were carried out to gain insights into the selectivity regulation and the overall reaction pathways over the Pd$_1$/ND@G and Pd$_n$/ND@G catalysts in BN hydrogenation reaction. According to EXAFS analysis, an isolated Pd atom, Pd$_1$-Graphene (Pd$_1$-Gr), and three-atom Pd cluster on graphene layer, Pd$_3$-Graphene, were constructed to represent the active sites on Pd$_1$/ND@G and Pd$_n$/ND@G catalysts, respectively. As shown in Fig. 4, H$_2$ gas adsorbs physically on Pd$_1$-Gr after the adsorption of BN molecule, which is exothermic by 0.38 eV. The activation of adsorbed H$_2$ carried a barrier of 1.15 eV and is endothermic by 0.61 eV, with one H atom adsorbed on the Pd single atom and one on C atom (the configuration is shown Supplementary Fig. 8), respectively. Next, the hydrogenation processes of BN undergo H transfer after the H$_2$ activation and C–H formation steps (as shown in Supplementary Fig. 8), which

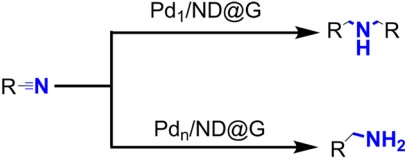

| Entry | Substrate | Pd$_1$/ND@G[a] | | | Pd$_n$/ND@G[b] | | |
|-------|-----------|------|----------|----------|------|----------|----------|
| | | t(h) | Con. (%) | Sel. (%) | t(h) | Con. (%) | Sel. (%) |
| 1 | | 8h | >99% | 98% | 0.5h | >99% | 98% |
| 2 | | 8h | >99% | 90% | 0.5h | >99% | 97% |
| 3 | | 9h | >99% | 90% | 1h | >99% | 98% |
| 4 | | 8h | >99% | 98% | 1h | >99% | 98% |
| 5 | | 10h | >99% | 85% | 2h | >99% | 90% |
| 6 | | 10h | >99% | 90% | 3h | >99% | 90% |
| 7 | | 8h | >99% | 91% | 3h | >99% | 98% |
| 8 | | 11h | >99% | 98% | 2h | >99% | 93% |

[a]Reaction condition: nitriles (0.5mmol), AB. (4mmol) and catalyst (30mg) in 10mL of CH$_3$OH were heated at 60℃. GC yield (%) were shown used m-Xylene as the internal standard.
[b]Reaction condition: nitriles (0.5mmol), AB. (3mmol) and catalyst (10mg) in 10mL of CH$_3$OH were heated at 40℃. GC yield (%) were shown used m-Xylene as the internal standard.

**Fig. 3 Substrate extension.** Catalytic performance of Pd$_1$/ND@G and Pd$_n$/ND@G catalyst in transfer hydrogenation of different substituted nitriles.

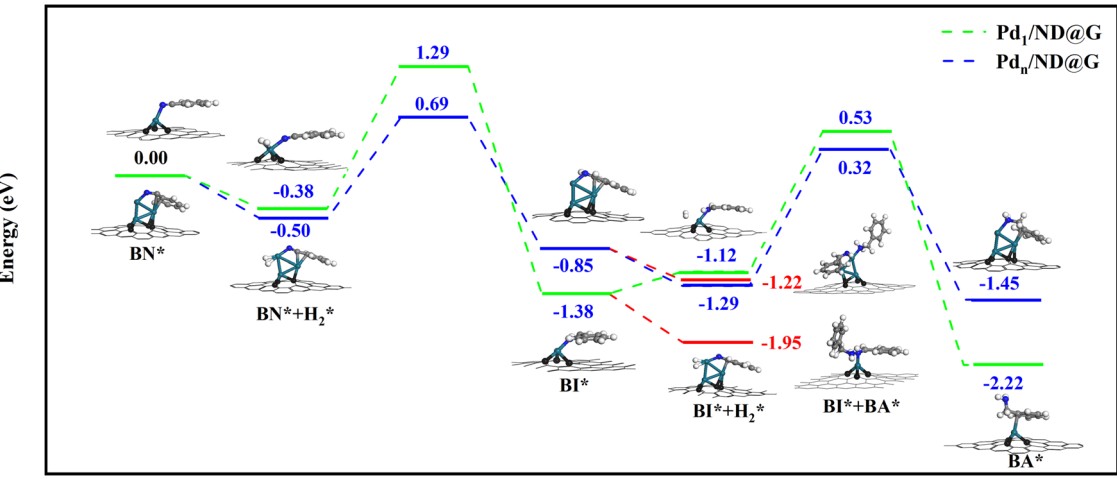

**Fig. 4 Step-by-step hydrogenation mechanism of benzonitrile to benzylamine on Pd1/ND@G and Pdn/ND@G.** Color code: Pd₁/ND@G (green line) and Pd$_n$/ND@G (blue line). Pd cyan, C gray, H white.

is exothermic by 1.0 eV and has an effective barrier of 1.67 eV. The further hydrogenation reaction from BI to BA is a similar process, which is exothermic by 1.1 eV and surfer from a higher effective barrier 1.91 eV. However, It should be noted that the dissociative adsorption process of the second H₂ is significantly harder thermodynamically and kinetically: the physical adsorption of the second H₂ gas is endothermic by 0.26 eV while that of the first H₂ is exothermic by 0.38 eV; The effective barrier of the second H₂ dissociative adsorption is 1.25 eV, slightly higher than 1.15 eV of the first H₂ gas. These calculated results indicate that the resident time of the BI surface intermediate is extended on Pd₁-Gr since the difficult activation of the second H₂, which helps the condensation reaction of BI surface intermediate by BA to form the *N*-benzylidenebenzylamine (DBI) intermediate. Then, DBI undergoes hydrogenation processes to generate the product (DBA) on Pd₁-Gr. In addition, as shown in Fig. 4 (red lines), we found that the further adsorption of BA after the formation of BI surface intermediate is exothermic by 0.57 eV, stronger than that of the second H₂ molecule (endothermic by 0.26 eV), which again suggests that the BI surface intermediates prefer to condensate with BA to generate the DBI intermediate than to hydrogenate to obtain BA. The calculated results of Pd₁-Gr explained well the excellent selectivity of DBA.

On Pd₃-Graphene (Pd₃-Gr), H₂ molecule easily dissociates after the adsorption of BN molecule, which is exothermic by 0.50 eV (see Supplementary Fig. 9). Then, the hydrogenation steps of BN to BI process with effective barrier 1.19 eV and is exothermic by 0.35 eV. As shown in Supplementary Fig. 9, we can see that the dissociative adsorption of the second H₂ gas occurs easily without barriers and is exothermic by 0.44 eV, which could boost the hydrogenation of BI intermediates and shorten the resident time of the BI surface intermediate. The following hydrogenation of BI to BA is exothermic by 0.16 eV and has a higher effective barrier of 1.54 eV. What is more, Fig. 4 shows the adsorption of BA after the formation of BI intermediates (−0.37 eV) is weaker than the dissociative adsorption of H₂ gas on Pd₃-Gr (−0.44 eV), suggesting that the hydrogenation reaction will be dominant compared to the condensation reaction of BI with BA. The theoretical studies of BN hydrogenation on Pd₃-Gr reveal that the high selectivity of BA originates from the facile activation of the H₂ molecules and the BA weak adsorption after the formation of the BI intermediate. These calculation results are consistent with our experimental results.

## Discussion

In summary, we investigated the structure–performance relationship at atomic scale for hydrogenation of nitriles by employing Pd₁/NDG with Pd single atoms and Pd$_n$/NDG with fully exposed Pd clusters as the catalyst. The secondary amines (>98% selectivity) and primary amines (>98% selectivity) are selectively generated under mild reaction conditions over Pd₁/NDG and Pd$_n$/NDG, respectively. Due to high utilization of Pd in these two atomically dispersed catalysts, excellent reactivity was achieved compared with other catalytic systems. DFT calculation reveals that the intermediate BI is easier to further hydrogenate to BA on the Pd$_n$/NDG catalyst. While on the Pd₁/NDG catalyst, BI is more inclined to undergo condensation reaction and continuing hydrogenation to obtain DBA. The selectivity regulation strategy established over these catalysts with atomic precision in structure will pave the way for the rational design and construction of the highly selective catalyst with fully metal utilization efficiency.

## Methods

**Materials**. Nanodiamond (ND) powders (99.9%) were purchased from Beijing Grish Hitech Co., and further purified by hydrochloric acid. Pd precursor (Pd(NO₃)₂ solution) was analytical regent and purchased from Alfa Aesar without further purification. Benzonitrile was purchased from Alfa Aesar. Ammonia borane was purchased from Macklin.

*Preparation of ND@G.* ND powder was calcined to obtain ND@G at 1100 °C. (condition: heating rate 5 °C min⁻¹ for 4 h under 100 mL min⁻¹ flowing Ar gas) and then naturally cooled to room temperature. The as-prepared products were further purified by hydrochloric acid for 24 h and then washed with DI water. Finally, the as-prepared ND@G support with diamond core and defective graphene shell was obtained after drying in vacuum at 60 °C for 24 h.

*Preparation of Pd$_n$/ND@G and Pd₁/ND@G.* First, 200 mg ND@G was dispersed into 30 mL deionized water in a 100 mL round-bottom flask, and the mixture was ultrasonically treated to obtain a homogeneous suspension. Then, the pH value of ND@G supports suspension was adjusted to about 10 by dropping 0.25 M Na₂CO₃ solution. Second, a certain amount of Pd(NO₃)₂ solution (containing 0.016 g mL⁻¹ Pd, from Alfa Aesar) was diluted into 4 mL water, and then the pH value of the solution was adjusted to neutral using 0.25 M Na₂CO₃. Subsequently, the pH neutral Pd solution was introduced immediately to carbon support suspension by drop-wise under magnetic stirring at 100 °C, and then kept stirring at 100 °C in oil bath for 1 hour. At the end, the mixture was cooled to room temperature, collected by filter, and washed several times with deionized water, until it was free of Na⁺ and CO₃²⁻. Afterwards, the powders were dried at 60 °C for 12 h. The as-prepared catalysts were reduced in H₂ (10 vol% H₂ in He, flow rate = 20 mL min⁻¹) at 200 °C for 1 h before the catalytic reaction.

**Characterizations**. HRTEM images were taken by a FEI Tecnai G2 F20 working at 200 kV. Atomic resolution STEM images were recorded by Cs-corrected cold field-emission. The XRD patterns of the nanocarbon supported Pd catalysts were collected by using an X-ray diffractometer (D/MAX-2400) using a Cu $K\alpha$ source at a scan rate of $2°\,min^{-1}$. XAFS measurements were carried out at the BL14W1 station in Shanghai Synchroton Radiation Facility (SSRF, 3.5 GeV, 250 mA in maximum, Si (311) double crystals)

*Typical procedure for the synthesis of DBA*. 0.5 mmol benzonitrile (from Alfa Aesar), 4 mmol AB (from Macklin), 1 mmol *m*-xylene (from SCR) as internal standard, 30 mg Pd$_1$/ND@G and 10 mL methanol were added to a sealed tube (50 mL) and heated at 60 °C for 8 h. After quenching, the mixture was analyzed by gas chromatography (GC) with the *m*-xylene as internal standard.

*Typical procedure for the synthesis of BA*. 0.5 mmol benzonitrile, 3 mmol AB, 1 mmol *m*-xylene as internal standard, 10 mg Pd$_n$/ND@G and 10 mL methanol were added to a sealed tube (50 mL) and heated at 40 °C for 0.5 h. After quenching, the mixture was analyzed by GC with the *m*-xylene as internal standard.

*Analytic methods*. The Agilent 7890A gas chromatography instrument, equipped with a crosslinked capillary (19091J-413: 325 °C: 30 m × 320 μm × 0.25 μm) and a flame ionization detector, was used for the products analysis. The analytic conditions were as follows: The flow rate of the N$_2$ carrier gas was 20 mL min$^{-1}$, and the injection port temperature was 260 °C. The GC oven temperature program was conducted as follows: The temperature program ranges from r.t. to 60 °C and held at 60 °C for 8.5 min. Then the temperature program ranges from 60 to 80 °C at a heating rate of 25 °C min$^{-1}$ and heat preservation at 60 °C for 4 min. And then temperature program ranges from 80 to 200 °C at a heating rate of 20 °C min$^{-1}$ then held at 200 °C for 5 min. The detector temperature was set to 280 °C. The content of each compound was determined based on the internal standard.

Benzonitrile conversion and selectivity of BA and DBA and TOF were calculated as the follows:

$$\text{Benzonitrile conversion : Conv.} = (\text{mol of the benzylamine})/(\text{mol of inlet benzonitrile}) \times 100\% \tag{1}$$

$$\text{Selectivity of benzylamine : Selectivity} = (\text{mol of benzylamine})/(\text{mol of converted benzonitrile}) \times 100\% \tag{2}$$

$$\text{Selectivity of dibenzylamine : Selectivity} = 2(\text{mol of dibenzylamine})/(\text{mol of converted benzonitrile}) \times 100\% \tag{3}$$

$$\text{TOFs : TOFs} = (\text{moles of converted nitrile})/(\text{moles of total noble metal atoms} \times \text{reacton time}) \tag{4}$$

**Computational details**. The spin-polarized calculations were performed by VASP code[41,42]. The projector augmented wave (PAW) method is adopted to describe interaction between electron and ion[43,44]. The generalized gradient approximation (GGA) and the Perdew–Burke–Ernzerhof functional (PBE)[45] was used for the exchange and correlation energies. An energy cutoff was set to 400 eV. The convergence tolerance for electronic and ionic steps was set to 10$^{-5}$ and 0.03 eV Å$^{-1}$ when optimizing the structure of bulk and surfaces. One carbon atom on the graphene layer (5 × 5 unit cells) was removed, and Pd$_3$ cluster and Pd single atom doped in the carbon defect was used as models based on the EXAFS data. To avoid interaction from adjacent cells, a 20-Å-thick vacuum layer is used. The 2 × 2 × 1 Monkhorst–Pack k-point is set for the models, and the Gaussian smearing method is used with $\sigma = 0.1$ eV. The potential configurations of adsorbed reactants and intermediates were screened to find the most stable one. The transition states (TS) searching method based on constrained scan[46] was conducted here. Frequency calculations were performed to verify TS with only on imaginary frequency. Moreover, TS structure with slight displacement towards vibration direction was further optimized in order to verify the identity of TS. Simplified model-based IGRRHO approaches were adopted to estimate the entropy contributions of adsorption and desorption process in the solution in the free energy calculations[47]. We assumed that the translational entropy is mostly responsible for the entropy contribution. The translational entropy of molecules at 40 °C (reaction condition were calculated using Gaussian 09 package (the IGRRHO is default)[48]. It was estimated that BA molecules and H$_2$ molecules lost 0.58 and 0.39 eV of entropic energy (T*S) in the adsorption, respectively.

## Data availability

The data supporting this article and other findings are available from the corresponding authors upon request. Source data are provided with this paper.

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

## Acknowledgements

This work was supported by the Ministry of Science and Technology (2016YFA0204100, 2017YFB0602200), the National Natural Science Foundation of China (21961160722, 22072162, 91845201, 21703261, 21725301, 91645115, and 21821004), the Liaoning Revitalization Talents Program XLYC1907055, Dalian National Lab for Clean Energy, DNL Cooperation Fund 202001 and the Sinopec China. N.W. hereby acknowledges the funding support from the Research Grants Council of Hong Kong (Project Nos. C6021-14E, N_HKUST624/19 and 16306818). The XAS experiments were conducted in Shanghai Synchrotron Radiation Facility (SSRF).

## Author contributions

H.L., D.M. and H.S. conceived the research. Z.L. and F.H. conducted material synthesis and carried out the catalytic performance test. M.P. conducted the X-ray absorption fine structure spectroscopic measurements and analyzed the data. Y.C. and X.W. performed the DFT calculations. X.C. and N.W. contributed to the aberration-corrected high-angle annular dark-field scanning transmission electron microscopy. L.W. and Z.H. performed some of the experiments. The manuscript was primarily written by, Z.L., F.H., H.S., H.J., D.X., H.L., and D.M. All authors contributed to discussions and manuscript review.

## Competing interests

The authors declare no competing interests.
