## [Peer Review File · Nature Communications]

Title: Tuning the selectivity of catalytic nitriles hydrogenation by structure regulation in atomically dispersed Pd catalystsREVIEWER COMMENTS

Reviewer #1 (Remarks to the Author):

This paper by Prof. Ma and co-workers reports a very interesting study of atomically dispersed Pd and small cluster Pd_n catalysts to enable hydrogenation of benzonitriles. The authors have synthesized and characterized the catalysts (Pd_n on graphene support), tested the catalytic performance of these catalysts by hydrogenating benzonitrile (BN) and related chemical species, and performed theoretical calculations by way of Density Functional Theory (DFT).

The work claims that the single-dispersed Pd sites are the cause of selective hydrogenation to secondary amines, and small Pd_n clusters are the cause for selective hydrogenation to primary amines.

The authors have no discussion or introduction on the well-established fact that Pd metal catalysts are excellent catalysts to hydrogenate selectively to tertiary amines which may be shifting the product distributions. There are countless works done for both experimental and computational studies to this end.

It would have been prudent to include tertiary amine characterization for both the experimental and computational components of this work. Also, long-term stability of the atomically Pd₁ and Pd_n dispersed catalyst would provide a more rigorous support of the claims. Lastly, comparison of the catalysts presented herein to larger Pd_n nanoparticles would provide a comparison across dispersion scales, allow for more complete analysis.

It is for these reasons that I cannot support the publication of this work in its current state.

Reviewer #2 (Remarks to the Author):

Liu et al report the use of single-atom and small Pd clusters for the hydrogenation of benzonitriles and substituted benzonitriles. A striking effect in catalyst selectivity, to DBA and BA, respectively, is observed depending on the catalyst structure (i.e. yields > 90 % in each case). The result is certainly intriguing, and could be added to the increasing list of examples where supported single metal atoms offer distinct catalytic characteristics compared to their clustered version. In this case, the cluster example, a “fully exposed” Pd cluster, shows almost quantitative selectivity to the primary amine without any sign of over-reduction to toluene. Although the results are interesting, many questions and a few concerns arise:

- The paper states that these highly dispersed catalysts provide “excellent reactivity compared with other catalytic systems”. However, no data is presented on a typical Pd/Al₂O₃ or Pd/C nanoparticle (of around 1-3 nm). The authors should collect their own data (in their setup) and compare with their less

standard catalysts (activity and selectivity profiles)

- It would be worth noting that DBA is typically an undesired byproduct of the BN to BA hydrogenation
- The two catalysts presented in this paper are tested at different experimental conditions (concentration of reactant and temperature). Comparison must be reported at identical reaction conditions to truly assess the role of the catalyst structure. Additional data at 40 and 60C for each catalyst should be provided.
- Literature on single metal and cluster catalysis is (probably unconsciously) biased. 19 through 27 references are all about Chinese groups (maybe one or two are joint efforts). While those are relevant pieces of work, they do not capture appropriately the richness and diversity of groups contributing to the field (a number of very notable ones are missing).
- The “fully exposed” Pd cluster is an intriguing system. A brief discussion that explains why a noble metal aggregates in the form of single-atoms and small clusters forming a 1.5 to 2.5 nm conglomerate (as shown in Fig 1c) would be helpful. The authors may have reported this more in detail in prior communications, but I think the general reader will appreciate to have a little more background on this matter.
- At what temperature (in H₂) does the “fully exposed” Pd cluster become a typical Pd nanoparticle? How does it behave afterwards in the particular test presented here?
- At what temperature (in H₂) does the single Pd catalyst become a metal clusters? How does it behave afterwards?
- Small clusters and single-atoms are often sensitive to reactions with the ambient. Catalyst were reduced at 200C after synthesis for this work. Were the catalysts exposed to the ambient (moisture and O₂) before running the batch experiments and the various characterization techniques? If not, please provide a more accurate procedure on how contact with the ambient was avoided. If yes, I would have serious concerns that the active sites may not be the same catalytic in H₂ (e.g. under reaction conditions) and after exposure to ambient O₂.
- Is there any post-mortem analysis of the catalysts (after reaction)? Is there any leaching of Pd into the solvent? Can the catalyst be re-used without activity loss?
- How was the identification of products in Fig 3 done? It might be convenient to isolate and characterize some of the compounds via C and N NMR, and report a few isolated yields.

In summary, the manuscript could be suitable for publication in Nat. Commun., in my opinion, but after a major revision to address the questions above.

Reviewer #3 (Remarks to the Author):

The catalytic performance in the transfer hydrogenation of nitriles with NH₃BH₃(AB) as the hydrogen donor was evaluated over ND@G supported single Pd atom (Pd₁) and Pd cluster (Pd_n) catalysts in this study. The authors found that secondary amines are preferentially obtained over Pd₁, whereas primary amines are selectively generated over Pd_n cluster. The authors performed DFT to elucidate the mechanism behind. I agree with this authors that separating the homogeneous catalysts from the product mixture is difficult/energy intensive and replacing it with heterogeneous catalysts is one way to

reduce the overall processing cost. I consider that tuning the selectivity of the reaction using metal support interaction is an attractive and straightforward way. Overall, it seems interesting to the readership of this journal, but in my opinion, it needs improvement in material characterizations and analyses.

Followings is my questions/comments. I hope this will help improve the quality of this authors' study to strength their claim before the publication for the readership of this journal.

1. The details of how ND@G was prepared needs to be mentioned so that other researcher performs same experiment to validate this study. What kind of nano diamond was used (specific surface area, Raman spectra, XRD etc) and what is the difference between real diamond and this nano diamond.
2. I suggest authors to show line profiles of diamond lattice in supplementary Fig.1 of HRTEM image of ND@G. The term diamond is awkward to me because some literatures just refer it graphite which exhibit quite similar XRD profiles shown in Fig. S2. Is this diamond-like carbon (DLC)? DLC has several crystal structures. Is this ND cubic? I suggest identifying the structure.
3. Also, I suggest author to show TEM images of pristine ND and compare the atomistic structure with annealed one (ND@G). In my opinion, TEM images in supplementary Fig.1 is not adequate to claim the formation of graphene. One way to prove it is showing much high-resolution images but not limited to this method probably. I'm wondering if the author could show more convincing evidences for formation of graphene.
4. By seeing the k³-weighted EXAFS, the quality of the measured spectra can be judged but not shown. I also suggest including details of analyses and parameters that the author assumed in fitting in supplementary information.
5. Judged by TEM in Fig.1b, there seems not only Pd single atom but also many Pd few atom clusters (2 to 4 atoms). I suggest authors to describe this for fair description.
6. The diameter of nanocluster in Fig. 1 d is about 1.7 nm or more and I can see 8 atoms from left edge of cluster to the right. Thus, it consists of more than 60 atoms in the first layer of the cluster. All clusters are round shape and thus I consider that cluster is neither single layer nor raft. In this situation, many atoms are hidden by topmost surface Pd atoms. How can the authors conclude that those large cluster are fully exposed? There seems many surface "unexposed atoms" and the term "fully exposed" is awkward for this catalyst.
7. Comparison of TOF with other literature will help reader to understand the performance of this catalysts. I suggest including such information and also describing details of how you calculate the TOF as same as conversion and selectivity in line 292.
8. According to the Method, high mass Pd on ND@G can be prepared by increasing the precursor concentration. If so, I suggest authors to investigate the selectivity for this high mass Pd in same experimental condition as a control to claim Pd₁ and Pd_n are unique and different from the bulk.
9. I suggest that authors explain and use the abbreviations when they first appear, for example, Turnover frequency (TOF). Please check followings abbreviations; GC (see line 277, 281, and 282), BID, Pd₁-Gr in main manuscript, TOL in supplementary,
10. In DFT, two models, Pd₁-Gr and Pd₃-Gr were compared. First of all, the ball stick model is too small to identify what is what. Second, the Pd_n cluster consists of more than 60 atoms even if I assume that cluster is single atom layer as I roughly estimated above. Thus, it seems unreasonable to calculate Pd₃-

Gr as a model for Pd_n cluster. I suggest authors to construct a model close to the observed/prepared cluster and discuss the result. Again, I can see some of Pd is not single, and two or more adjacent atoms can be seen in TEM. Is this the reason why the selectivity is not 100 %? Finally, there seems no discussion on local charge for the models. The authors performed XPS for the prepared samples and thus I suggest authors to calculate Bader charges and discuss local charge.

11. M-Xylene is used as internal standard but is it stable to the catalysis in this study?

12. I suggest the authors to include information where they purchased for the all chemicals used in this study.

REVIEWER COMMENTS

Reviewer #1 (Remarks to the Author):

This paper by Prof. Ma and co-workers reports a very interesting study of atomically dispersed Pd and small cluster Pd_n catalysts to enable hydrogenation of benzonitriles. The authors have synthesized and characterized the catalysts (Pd_n on graphene support), tested the catalytic performance of these catalysts by hydrogenating benzonitrile (BN) and related chemical species, and performed theoretical calculations by way of Density Functional Theory (DFT).

The work claims that the single-dispersed Pd sites are the cause of selective hydrogenation to secondary amines, and small Pd_n clusters are the cause for selective hydrogenation to primary amines.

Comments: The authors have no discussion or introduction on the well-established fact that Pd metal catalysts are excellent catalysts to hydrogenate selectively to tertiary amines which may be shifting the product distributions. There are countless works done for both experimental and computational studies to this end.

Response: We appreciate the reviewer for very nice comment. The reduction of nitriles to amines is a well-known transformation reaction. However, the product selectivity is affected by many factors (catalysts, solvents, substrates, additives, etc.). Michael P. Haaf's group reported the efficient syntheses of tertiary alkyl amines from their corresponding alkyl nitriles in the presence of a heterogeneous palladium catalyst and a source of dihydrogen in aprotic solvents (*Tetrahedron Lett.* **2012**, 53, 4426–4428). And Yasunari Monguchi et al. reported hydrogenation of aliphatic nitriles in cyclohexane efficiently proceeded at 25–60 °C under ordinary hydrogen gas pressure to afford the corresponding tertiary amines in the presence of palladium on carbon (Pd/C) as a catalyst (*J. Org. Chem.* **2017**, 82, 10939–10944). Therefore, the formation of tertiary amines generally depends on the structure of substrates, and it can be noted that the hydrogenation of aliphatic nitriles often yields tertiary amines.

However, as to the aromatic nitriles, we can see from the numerous reports, many primary amines and secondary amines were obtained by employing Pd-based catalysts (*Appl. Catal. A* **2008**, 349, 40–45, *Catal. Sci. Technol.*, **2019**, 9, 2266–2272, *Catal. Sci. Technol.*, **2014**, 4, 629–632, *J. Catal.* **2010**, 274, 176–191, *Catal. Commun.* **2012**, 28, 9–12). Lu's group also demonstrated that isolating Pd with Ni breaks the strong metal-selectivity relations in hydrogenation of nitriles, and prompts the yield of secondary amines (*Nat. Commun.* **2019**, 10, 4998).

Comments: It would have been prudent to include tertiary amine characterization for both the experimental and computational components of this work. Also, long-term stability of the atomically Pd₁ and Pd_n dispersed catalyst would provide a more rigorous support of the claims. Lastly, comparison of the catalysts presented herein to larger Pd_n nanoparticles would provide a comparison across dispersion scales, allow for more complete analysis.

Response: We appreciate the reviewer for the nice suggestion. We explored the stability of Pd₁/ND@G and Pd_n/ND@G catalyst with the overnight hydrogenation of BN. As shown in Figure R1a, the dibenzylamine (DBA) yield of dibenzylamine decreased slightly, a small amount of dibenzylamine is converted to benzylamine and no tertiary amine was detected. Notably, the selectivity of the products did not change when the reaction time was extended on the Pd_n/ND@G

catalyst (Figure R1b). Because of the limitation of our computing resources for such large molecules (Tribenzylamine), the production of tertiary amines was not calculated. Meanwhile, the commercial Pd/C (5 wt.%) catalyst was employed as a reference, although the Pd/C catalyst exhibited higher selectivity of benzylamine (BA), its catalyst activity was quite worse compared with that of the fully exposed Pd_n/ND@G catalyst (Figure R1c).

Figure R1. (a) Time-conversion plot for production formation from the transfer hydrogenation of benzonitrile over Pd₁/ND@G. Reaction conditions: solvent, methanol, 10 mL; BN, 0.5 mmol; catalyst, 30 mg; AB, 4 mmol; Temperature, 60 °C. (b) Time-conversion plot for production formation from the transfer hydrogenation of benzonitrile over Pd_n/ND@G. Reaction conditions: solvent, methanol, 10 mL; BN, 0.5 mmol; catalyst, 10 mg; AB, 3 mmol; Temperature, 40 °C. (c) Product yield for transfer hydrogenation of benzonitrile over ND@G, Pd₁/ND@G (Time, 8 h) and Pd_n/ND@G (Time, 30 min), Pd/C (Time, 30 min).

Special thanks to the reviewer for his/her insight comments, which really help us a lot in improving the manuscript.

Reviewer #2 (Remarks to the Author):

Liu et al report the use of single-atom and small Pd clusters for the hydrogenation of benzonitriles and substituted benzonitriles. A striking effect in catalyst selectivity, to DBA and BA, respectively, is observed depending on the catalyst structure (i.e. yields > 90 % in each case). The result is certainly intriguing, and could be added to the increasing list of examples where supported single metal atoms offer distinct catalytic characteristics compared to their clustered version. In this case, the cluster example, a “fully exposed” Pd cluster, shows almost quantitative selectivity to the primary amine without any sign of over-reduction to toluene. Although the results are interesting, many questions and a few concerns arise:

Comments: The paper states that these highly dispersed catalysts provide “excellent reactivity compared with other catalytic systems”. However, no data is presented on a typical Pd/Al₂O₃ or Pd/C nanoparticle (of around 1-3 nm). The authors should collect their own data (in their setup) and compare with their less standard catalysts (activity and selectivity profiles)

Response: We thank the reviewer for the good point. As shown in Figure R2, the commercial Pd/C (5 wt.%) catalyst was used as a reference, although the Pd/C catalyst exhibited higher selectivity of benzylamine (BA), its catalyst activity was quite worse compared with that of the fully exposed Pd_n/ND@G catalyst.

Figure R2. Product yield for transfer hydrogenation of benzonitrile over ND@G, Pd₁/ND@G (Time, 8 h) and Pd_n/ND@G (Time, 30 min), Pd/C (Time, 30 min).

Comments: It would be worth noting that DBA is typically an undesired byproduct of the BN to BA hydrogenation.

Response: We appreciate the reviewer for the very nice comment. In this paper, we investigated the structure-performance relationship at atomic scale for hydrogenation of nitriles by employing Pd₁/NDG with Pd single atoms and Pd_n/NDG with fully exposed Pd clusters as the catalyst. The secondary amines and primary amines are selectively generated over Pd₁/NDG and Pd_n/NDG, respectively. It is worth noting that a few single atoms inevitably exist in the cluster catalysts, and these single atoms affect the selectivity of the products. However, it is still the clusters that play a major role in the Pd_n/NDG catalyst. And this is probably further proof it's easier to get a DBA over single atoms catalyst in the hydrogenation of BN.

Comments: The two catalysts presented in this paper are tested at different experimental conditions (concentration of reactant and temperature). Comparison must be reported at identical reaction conditions to truly assess the role of the catalyst structure. Additional data at 40 and 60C for each catalyst should be provided.

Response: We thank the reviewer for the comment. Different catalysts may have different suitable reaction conditions. We determine the final optimal reaction conditions (concentration of reactant and temperature) through optimization. Additional data at 40°C and 60°C for each catalyst has been added to Supporting Information (**Table S3 and S5**). In addition, Liu group's describe a selective cobalt-catalyzed chemodivergent transfer hydrogenation of nitriles to synthesize primary, secondary, and tertiary amines. The two catalysts were also tested at different reaction temperatures (*Angew. Chem. Int. Ed.* **2016**, 55, 14653).

Comments: Literature on single metal and cluster catalysis is (probably unconsciously) biased. 19 through 27 references are all about Chinese groups (maybe one or two are joint efforts). While those are relevant pieces of work, they do not capture appropriately the richness and diversity of groups contributing to the field (a number of very notable ones are missing).

Response: We thank the reviewer for the nice suggestion. We're terribly sorry for missing some good studies. We have adjusted some of the relevant references to help readers better understand the relevant concepts. References: (*Angew. Chem., Int. Ed.* **2015**, 54, 11265–11269; *ACS Catal.* **2017**, 7, 3147–3151; *Catal. Sci. Technol.* **2016**, 6, 4081–4085; *ACS Catal.* **2013**, 3, 2449–2455).

Comments: The “fully exposed” Pd cluster is an intriguing system. A brief discussion that explains why a noble metal aggregates in the form of single-atoms and small clusters forming a 1.5 to 2.5 nm conglomerate (as shown in Fig 1c) would be helpful. The authors may have reported this more in detail in prior communications, but I think the general reader will appreciate to have a little more background on this matter.

Response: We appreciate the reviewer for his/her nice comments which inspired us to add corresponding discussion of the “fully exposed” clusters. Following the reviewer's advice, we added a brief discussion in the revised manuscript (see Page 5 Line 2) (FECCs offer diverse surface sites formed by an ensemble of metal atoms, comparing with single-atom catalyst, it not only provides maximized atom utilization but also possesses rich active sites and easily identified coordination structures. FECC is so highly dispersed that all the metal atoms within it are available for the adsorption and transformation of reactants. As its stable metal loading can be higher than that of SAC, the FECC usually exhibits higher mass specific activity than the SAC, which is critically important for industrial applications). And detailed and original description of clusters can be found in our previous studies (*ACS Cent. Sci.* **2021**, 7, 262–273, *ACS Catal.* **2019**, 9, 5998–6005, *Nat. Commun.* **2021**, 12, 2664-2664.).

Comments: At what temperature (in H₂) does the “fully exposed” Pd cluster become a typical Pd nanoparticle? How does it behave afterwards in the particular test presented here? At what temperature (in H₂) does the single Pd catalyst become a metal clusters? How does it behave

afterwards?

Response: We thank the reviewer for the comments. Our Pd-based precursor catalysts were reduced at 200°C, Actually, we have investigated the thermal stability of Pd₁/ND@G catalyst, the result demonstrated that atomically dispersed Pd on ND@G could be stable at 350 °C in hydrogen containing atmospheres (Figure R3a). At 500 °C or higher, some of isolated Pd atoms started to transform into Pd clusters, but a large proportion of Pd atoms still keep their isolated nature (see Figure R3b). On the change of metal cluster catalyst, the stability of cluster catalyst in the hydrogen containing atmospheres (350 °C) was investigated. As shown in Figure R3c and R3d, Pd₃/ND@G still maintains the fully exposed cluster structure at 350 °C for 5h. In this paper about the liquid phase reaction at a lower temperature, the structure of the catalyst will not be affected by temperature changes, catalyst performance is always consistent. So, we did not further study the effect of the corresponding catalyst changes on the reaction.

Figure R3. AC-HAADF-STEM image (a) Pd₁/ND@G-350°C and (b) of Pd₁/ND@G-500°C. The AC-HAADF-STEM images of the Pd₃/ND@G -350°C.

Comments: Small clusters and single-atoms are often sensitive to reactions with the ambient. Catalyst were reduced at 200C after synthesis for this work. Were the catalysts exposed to the ambient (moisture and O₂) before running the batch experiments and the various characterization techniques? If not, please provide a more accurate procedure on how contact with the ambient was avoided. If yes, I would have serious concerns that the active sites may not be the same catalytic in H₂ (e.g. under reaction conditions) and after exposure to ambient O₂.

Response: Thanks for the careful review. Our catalyst is really exposed to the ambient (moisture and O₂) before running the batch experiments and the various characterization techniques. The as-prepared catalysts were reduced in H₂ (10 vol% H₂ in He, flow rate= 20 mL min⁻¹) at 200 °C for 1h before the catalytic reaction and various characterization. Through the characterization (XAFS and XPS) of the reduced catalyst, it is found that there is stronger charge transfer between the metal Pd and the support. Notably, the Pd valence state of Pd_n/ND@G and Pd₁/ND@G are both positively charged and the Pd metal appears as an oxidation state. So our catalyst is stable and is not greatly

affected by other factors in the ambient (moisture and O₂) environment. From what has been discussed above, the active sites are the same under catalytic reaction conditions (low temperature, less hydrogen) and after exposure to ambient O₂.

Comments: Is there any post-mortem analysis of the catalysts (after reaction)? Is there any leaching of Pd into the solvent? Can the catalyst be re-used without activity loss?

Response: The two catalysts after reaction were characterized by AC-HAADF-STEM. In the Pd₁/ND@G catalyst after reaction (Figure R4a, b), the leaching of a few Pd atoms leads to the decrease of the density of metal Pd, and then the activity of the Pd₁/ND@G catalyst was partially reduced. However, the Pd₁/ND@G catalyst still retains good selectivity of dibenzylamine (Figure R4e) and suggests that the Pd single atoms still play a major role. As shown in Figure R4f, there was no significant change in activity of Pd_n/ND@G catalyst after reaction, but the selectivity of dibenzylamine increased slightly. This may be related to the decrease in the number of clusters and increase in the number of single atoms caused by the leaching of Pd in the Pd₁/ND@G catalyst (Figure R4c, d).

Figure R4. The AC-HAADF-STEM images of the Pd₁/ND@G catalyst at low (a) and high (b) magnification after the transfer hydrogenation of benzonitrile. The AC-HAADF-STEM images of the Pd_n/ND@G catalyst at low (c) and high (d) magnification after the transfer hydrogenation of benzonitrile. (e) The recycling experiments of the Pd₁/ND@G catalyst used for the transfer hydrogenation of benzonitrile. Reaction conditions: solvent, methanol, 10 mL; BN, 0.5 mmol; catalyst, 60 mg; AB, 4 mmol; Temperature, 60 °C; Time, 8 h. (f) The recycling experiments of the Pd_n/ND@G catalyst used for the transfer hydrogenation of benzonitrile. Reaction conditions: solvent, methanol, 10 mL; BN, 0.5 mmol; catalyst, 10 mg; AB, 3mmol; Temperature, 40 °C; Time, 30 min.

Comments: How was the identification of products in Fig3 done? It might be convenient to isolate and characterize some of the compounds via C and N NMR, and report a few isolated yields.

Response: We thank the reviewer for pointing this out for us. All product residues were purified respectively by chromatography on silica gel, and isolated yields of products were reported in SI. The products were confirmed by NMR spectra. As shown in Figure R5, we take (3,4-difluorophenyl) methanamine and bis (3,4-difluorobenzyl) amine for example, it can be found that the ^{15}N spectrum signal of the primary amine is slightly higher than that of the secondary amine, but the peak intensity of both products is poor, and it is difficult and inaccurate to confirm products structure by ^{15}N spectrum. Therefore we finally confirmed the product by ^1H and ^{13}C NMR spectra. Generally, related products have been confirmed by ^1H and ^{13}C NMR spectra in many references (Angew. Chem. Int. Ed. **2016**, 55, 14653–14657, ACS Catal. **2018**, 8, 9125-9130, J. Am. Chem. Soc. **2017**, 139, 13554-13561, Chem Commun. **2014**, 50, 3512-3515, **2011**, Chem. Eur. J. 17, 13308-13317). And we further supplemented GC-MS to analyze the structure of the products. The related NMR spectra and GC-MS results are placed in SI.

Figure R5. ^{15}N spectrum of (3,4-difluorophenyl)methanamine and bis(3,4-difluorobenzyl)amine

Special thanks to the reviewer for his/her insight comments, which really help us a lot in improving the manuscript.

Reviewer #3 (Remarks to the Author):

The catalytic performance in the transfer hydrogenation of nitriles with $\text{NH}_3\text{BH}_3(\text{AB})$ as the hydrogen donor was evaluated over ND@G supported single Pd atom (Pd1) and Pd cluster (Pdn) catalysts in this study. The authors found that secondary amines are preferentially obtained over Pd1, whereas primary amines are selectively generated over Pdn cluster. The authors performed DFT to elucidate the mechanism behind. I agree with this authors that separating the homogeneous catalysts from the product mixture is difficult/energy intensive and replacing it with heterogeneous catalysts is one way to reduce the overall processing cost. I consider that tuning the selectivity of the reaction using metal support interaction is an attractive and straightforward way. Overall, it seems interesting to the readership of this journal, but in my opinion, it needs improvement in material characterizations and analyses.

Followings is my questions/comments. I hope this will help improve the quality of this authors' study to strength their claim before the publication for the readership of this journal.

Comments: The details of how ND@G was prepared needs to be mentioned so that other researcher performs same experiment to validate this study. What kind of nano diamond was used (specific surface area, Raman spectra, XRD etc) and what is the difference between real diamond and this nano diamond.

Response: We thank the reviewer for the nice suggestion. We are sorry for not clearly describing the preparation method. We have added the detailed description in the experimental section. ND powders (high-purity grade) were bought from Beijing Grish Hitech, P.R. China, and were synthesized by the detonation explosive method followed by acid washing for purification. About the characterization of ND, specific surface area, Raman spectra, XRD, HRTEM images, DRIFT spectra and XPS were presented in our group's previous research (Chem. Eur. J. **2014**, 20, 6324 – 6331). GRISH detonation Nano-diamond (DND), also called ultra-fine diamond (UFD), is achieved from the dissociative carbon in super high pressure and temperature during the detonation by the oxygen-negative explosive. It is different from the synthesized diamond, the shape of DND is sphere without sharp edges and the crystal size is only about 4 nm.

Comments: I suggest authors to show line profiles of diamond lattice in supplementary Fig.1 of HRTEM image of ND@G. The term diamond is awkward to me because some literatures just refer it graphite which exhibit quite similar XRD profiles shown in Fig. S2. Is this diamond-like carbon (DLC)? DLC has several crystal structures. Is this ND cubic? I suggest identifying the structure.

Response: We thank the reviewer for the suggestion. The ND@G support is composed of a nano-diamond core and a defective, ultrathin graphene nanoshell. The core of each particle consists of a well-defined crystalline diamond with (111) planes covered by an outer shell of unstructured amorphous carbon. With increasing annealing temperature, the particles gradually become more highly ordered. In fact, our group has done a lot of work on the proof of graphene in previous reports (Chem. Eur. J. **2014**, 20, 6324 – 6331, ACS Catal. **2017**, 7, 3349–3355, J. Am. Chem. Soc. **2018**, 140, 13142–13146, Nat. Commun. **2019**, 10:4431, ACS Catal. **2019**, 9, 5998–6005). In this paper, we are sorry that we provided readers a little bit of information about the ND@G support but we've focused on catalytic reactions. We have added relevant citations to help readers better understand our work.

Comments: Also, I suggest author to show TEM images of pristine ND and compare the atomistic

structure with annealed one (ND@G). In my opinion, TEM images in supplementary Fig.1 is not adequate to claim the formation of graphene. One way to prove it is showing much high-resolution images but not limited to this method probably. I'm wondering if the author could show more convincing evidences for formation of graphene.

Response: We thank the reviewer for pointing this out for us. About the characterization of pristine ND and the atomistic structure with annealed one (ND@G) TEM comparison but not limited to this one characterization (Figure R6, Table R1) were described detailedly in our previous paper (Chem. Eur. J. 2014, 20, 6324 – 6331).

Figure R4. a) Typical HRTEM images of pristine ND, b) Raman spectra of ND-derived samples. c) XRD patterns of ND-derived samples. d) DRIFT spectra of ND-derived samples during annealing. e) Electron energy loss spectroscopy (EELS) profiles of ND derived samples. (Chem. Eur. J. 2014, 20, 6324 – 6331).

Table R1. Physicochemical characterization data of ND-derived samples. (Chem. Eur. J. 2014, 20, 6324 – 6331).

Sample	Sp ² (%) ^[a]
ND	27
ND-550	28
ND-800	32
ND-1000	51
ND-1100	52
ND-1300	71

[a] The fraction of sp² carbon estimated from EELS spectra.

Comments: By seeing the k3-weighted EXAFS, the quality of the measured spectra can be judged but not shown. I also suggest including details of analyses and parameters that the author assumed in fitting in supplementary information.

Response: Thanks for the good suggestion. X-ray absorption spectra measurements were performed at the BL14W1 beamline in Shanghai Synchrotron Radiation Facility (SSRF, Energy 3.5 GeV, Current 250 mA in maximum, Si (311) double-crystals as double crystal monochromator which could cover the photon energy range from 8.5 keV to 50 keV). The Palladium K-edge (24,350 eV) of the samples was measured in fluorescence mode, using a Lytle detector to collect the data. Pd foil and PdO were used as standard references and measured in transmission mode with Oxford ion chamber as detector. The XAFS samples were sealed in Kapton films with Ar protection after activation, and the whole process was performed in a glovebox. All XAFS spectra were analyzed using the Ifeffit package version 1.2.11. The extended XAFS oscillation was fitted according to a back-scattering equation, using FEFF models generated from the standard structures Pd(Fm3m) and PdO(P42/mmc) to extract the standard scattering path of Pd-Pd and Pd-C/O, which was used to determine the coordination number. The wavelet transform was processed using MATLAB 2020b. The k-space data of each sample of exact k range of Fourier transformation was processed with the script, generating a matrix containing the continuous wavelet transform data. The relevant fitting parameters are listed in the table R2.

Table R2. Structural parameters extracted from quantitative EXAFS curve-fitting.

Sample	Shell	C.N. ^[a]	R(Å) ^[b]	$\Delta\sigma^2$ ($/10^{-3} \text{ \AA}^2$) ^[c]	ΔE_0 (eV) ^[d]	R factor
Pd foil	Pd-Pd	12.0	2.74	5.86	-6.49	0.001
PdO	Pd-O	4.0	2.01	2.13	-2.62	0.001
Pd ₁ /ND@G	Pd-Pd	0	-	-	-	0.027
	Pd-C/O	2.6	2.04	1.80	4.660	
Pd _n /ND@G	Pd-Pd	1.9	2.75	5.76	8.603	0.030
	Pd-C/O	2.5	2.02	5.19		

^[a] C.N. is the coordination number. ^[b] R is interatomic distance (the bond length between Pd central atoms and surrounding coordination atoms). ^[c] σ^2 is Debye-Waller factor (a measure of thermal and static disorder in absorber scatterer distances). ^[d] ΔE_0 is edge energy shift (the difference between the zero kinetic energy value of the sample and that of the theoretical model).

Comments: Judged by TEM in Fig.1b, there seems not only Pd single atom but also many Pd few atom clusters (2 to 4 atoms). I suggest authors to describe this for fair description.

Response: We appreciate the reviewer for the suggestion. Indeed, Because of some uncontrollable factors in the preparation process, it is inevitable that there are a few of close Pd metal atoms in the single atoms catalysts. However, most of them are still in the form of single atoms, and no Pd-Pd was observed according to the EXAFS results (Table R2), their catalytic properties are not greatly affected. We modified the relevant description according to the suggestions of reviewers (see Page 7 Line 11).

Comments: The diameter of nanocluster in Fig. 1 d is about 1.7 nm or more and I can see 8 atoms from left edge of cluster to the right. Thus, it consists of more than 60 atoms in the first layer of the cluster. All clusters are round shape and thus I consider that cluster is neither single layer nor raft. In this situation, many atoms are hidden by topmost surface Pd atoms. How can the authors conclude that those large cluster are fully exposed? There seems many surface “unexposed atoms” and the term “fully exposed” is awkward for this catalyst.

Response: We appreciate the reviewer for very nice comment. In fact, the regions in Fig. 1d were island-like aggregates (~1.7nm) composed of multiple small clusters. In our previous work (*ACS Catal.* **2019**, 9, 5998; *Nat. Commun.* **2021**, 12, 2664), as shown in Figure R7, the atomic model (a) showing a Pt₃ on defective graphene surface and its projection view (b) along the electron beam direction. The simulated STEM image according to the atomic model was shown in Figure R7c, where carbon atoms are invisible due to the low contrast. Clearly, the optimized structure of the Pt₃ cluster was not in parallel with carbon support surface, which may explain the irregular atomic structures of Pt clusters under STEM due to the projection nature of transmission electron microscopy technique. And the surface of the carrier ND@G is curved and irregular, island-like aggregates of large clusters contain smaller clusters at different locations. The different height positions in the sample show a distinct color difference on the transmission electron microscope, so that part of the metal appears to be unexposed (as shown in Figure R8). On the other hand, the EXAFS results (Table R2) also indicate that most of the Pd clusters were fully exposed.

Figure R7. HAADF-STEM image simulations. Atomic model (a) showing a Pt₃ on defective graphene surface and its projection view (b) along the electron beam direction. (c) Simulated image of the structure in (b) using experimental parameters, where carbon atoms are invisible due to the low contrast (*ACS Catal.* 2019, 9, 5998).

Figure R8. HAADF-STEM imaging diagram of Pd_n/ND@G

Comments: Comparison of TOF with other literature will help reader to understand the performance of this catalysts. I suggest including such information and also describing details of how you calculate the TOF as same as conversion and selectivity in line 292.

Response: Thanks for the good suggestion. We added the relevant description according to the suggestions of reviewers.

$$\text{TOFs} = \frac{\text{moles of nitrile converted}}{\text{moles of total noble metal atoms} \times \text{reaction time}}$$

Comments: According to the Method, high mass Pd on ND@G can be prepared by increasing the precursor concentration. If so, I suggest authors to investigate the selectivity for this high mass Pd in same experimental condition as a control to claim Pd₁ and Pd_n are unique and different from the

bulk.

Response: We thank the reviewer for the nice suggestion. We chose Pd/C (5 wt %) catalyst as a comparison to investigate the catalytic performance of the bulk catalysts (Figure R9). Although Pd/C catalyst exhibited higher selectivity of benzylamine (BA), its catalyst activity was quite worse compared with the fully exposed Pd_n/ND@G catalyst.

Figure R9. Product yield for transfer hydrogenation of benzonitrile over ND@G, Pd₁/ND@G (Time, 8 h) and Pd_n/ND@G (Time, 30 min), Pd/C(Time, 30 min).

Comments: I suggest that authors explain and use the abbreviations when they first appear, for example, Turnover frequency (TOF). Please check following abbreviations; GC (see line 277, 281, and 282), DBI, Pd1-Gr in main manuscript, TOL in supplementary,

Response: We appreciate the reviewer for the suggestion. We explained and used the abbreviations when they first appear in main manuscript and supplementary.

Comments: In DFT, two models, Pd1-Gr and Pd3-Gr were compared. First of all, the ball stick model is too small to identify what is what. Second, the Pd_n cluster consists of more than 60 atoms even if I assume that cluster is single atom layer as I roughly estimated above. Thus, it seems unreasonable to calculate Pd3-Gr as a model for Pd_n cluster. I suggest authors to construct a model close to the observed/prepared cluster and discuss the result. Again, I can see some of Pd is not single, and two or more adjacent atoms can be seen in TEM. Is this the reason why the selectivity is not 100 %? Finally, there seems no discussion on local charge for the models. The authors performed XPS for the prepared samples and thus I suggest authors to calculate Bader charges and discuss local charge.

Response: Thanks for the good suggestion. We resized the ball-and-stick model so that the reader could identify them clearly. By responding to comment 6, we explain the related electron microscopy characteristics of our fully exposed cluster catalyst. Although the electron microscopy information is not very clear, we mainly rely on the analysis of EXAFS data (Table R2) to establish our catalyst model. As mentioned in comment 5, it is inevitable that there are a small number of close Pd metal atoms in the single atoms catalysts. The product selectivity is not 100% due to a small amount of other forms of Pd metal in the single atoms catalyst.

Table R3. DFT calculation parameters of Pd bader charges in the Pd₁/ND@G and Pd_n/ND@G.

Sample	Q(Pd)(e)
 Pd ₁ /ND@G	0.45
 Pd _n /ND@G	1—0.41 2—0.25 3—-0.27

According to the Bader charge analysis, the local charge on the Pd species is 0.45 e in the Pd₁/ND@G coordination, which is higher than that in the Pd_n/ND@G coordination (0.41 e, 0.25 e, -0.27 e), illustrating the higher positively charged of the Pd species in the Pd₁/ND@G. This result is consistent with the conclusion drawn from Fig. 1e (XANES) and Fig.S3(XPS), further implying a stronger charge transfer between Pd species and the ND@G support in the Pd₁/ND@G catalyst.

Comments: M-Xylene is used as internal standard but is it stable to the catalysis in this study?

Response: We thank the reviewer for the comments. The selection of internal standard (M-Xylene) is determined by our compliance with the selection requirements of internal standard (a. The original sample did not contain components. b. The retention time should be close to that of the object to be measured, but not overlapping. c. It is a standard substance of high purity, or a substance with known content. d. It has certain chemical stability under given chromatographic conditions.). We strictly observe these rules, M-Xylene is used as internal standard and it is stable to the catalysis in this study.

Comments: I suggest the authors to include information where they purchased for the all chemicals used in this study.

Response: Thanks for the good suggestion. We add the purchase information of the all chemicals to the later part.

Special thanks to the reviewer for his/her insight comments, which really help us a lot in improving the manuscript.

REVIEWER COMMENTS

Reviewer #2 (Remarks to the Author):

I appreciate the author's comments to address the various questions by the Reviewers. Although there is an honest effort to answer the points made, I find the behavior of the "fully exposed" Pd clusters fundamentally analogous to that of conventional Pd/C for the chosen chemistry. While it is true that higher dispersion may be boosting the catalyst activity (on a per total Pd basis), the difference is, in my opinion, insufficient to justify publication of these results in such a highly prestigious journal like Nature Communications. I consider this point critical because previous releases by the same group already disclose important synthesis and structural properties of the catalyst, and the current paper fails to demonstrate a genuinely distinct catalytic performance when compared to other alternatives. I find the work more suitable for publication as a full Research Article in JACS or ACS Catalysis.

Reviewer #3 (Remarks to the Author):

Details of how ND@G was prepared is included in the revised manuscript but still I do not see line profiles of ND@G. I understood that the authors studied ND before using XRD, Raman, but I do not understand why these authors cannot show the line profiles for lattice fringes and local diffraction pattern in this study?

Author explained that the 1.7 nm grain is formed due to the electron exposure in the comment this reviewer. If so, the grains circled by red shown in Fig. 1d are not the true structure of Pd_n/ND/@G but it is the one transformed after the exposure of electron. I understood the image shown in Fig. 1d is not the pristine Pd clusters through the authors' reply but without such explanation, it is not accurate and confusing.

This arises another question. Why single Pd atoms were able to image as single (Fig. 1b)? If the agglomeration occurs for Pd₃/ND@G, there would be no reason that same agglomeration occurs for Pd₁/ND@G.

Authors mentioned "Although the electron microscopy information is not very clear, we mainly rely on the analysis of EXAFS data (Table R2) to establish our catalyst model." To me, it is hard to agree with this because TEM would give plenty information regarding local atomic structure as this author also showed an atomic resolution TEM (but no line profiles for SACs and the diffraction for ND, which I still consider important to be included). Together with EXAFS, existence of the single atom can be supported but needs careful evaluations.

Appl. Phys. Lett. 116, 191903 (2020); <https://doi.org/10.1063/5.0008748>

Notably, EXAFS is not sensitive to highly dispersed metal oxide species with particle sizes <1 nm.

Nature Catalysis volume 4, pages453–456 (2021)

Relying on the analysis of EXAFS alone would mislead the reader and this is the current problem of the research community. Besides, the suggested k³-weighted EXAFS to show the quality of the measurement is not shown yet, and parameters assumed for example,

I appreciate the author to prepare the high mass loading Pd (5wt% one) and performed catalysis in the response to this reviewer, but again it is not characterized well, by for example, high resolution TEM, XRD.

TOF is sensitive to the temperature and please provide the information in the vicinity of description of TOF, for example, TOF@300K, otherwise reader needs to look up this information.

I thank the authors effort to take this reviewer's comment account in their work, but I still have strong concerns regarding the quality of material characterization and rationale based on DFT considerations. (there is still large mismatch between observed 1.7 nm Pd NP with Pd₃/ND@G), and have anxious that this work mislead the community. Pd_n/ND@G (blue line) needs to be expressed as Pd₃/ND@G (blue line) since the authors calculate for the latter specific model. I recommend the authors to provide more concrete data for the materials used (Pd₁, Pd₃, and Pd_n made by high concentration precursor.). In the current version, while this reviewer finds great interest especially the origin of observed selectivity, the provided data is not convincing in my opinion. The other concern is catalysis test condition, which are different as pointed out by the other reviewer. I believe when TOF or conversion efficiency are compared, the system needs to be kept the same for fair comparison and thus, unfortunately the current work needs significant improvements, which impressed me the current work is premature. I believe that it is necessary for the structure to be well analyzed further and its structure and performance to be compared on a one-to-one basis under the same catalysis conditions.

Reviewer #2 (Remarks to the Author):

Comments: I appreciate the author's comments to address the various questions by the Reviewers. Although there is an honest effort to answer the points made, I find the behavior of the "fully exposed" Pd clusters fundamentally analogous to that of conventional Pd/C for the chosen chemistry. While it is true that higher dispersion may be boosting the catalyst activity (on a per total Pd basis), the difference is, in my opinion, insufficient to justify publication of these results in such a highly prestigious journal like Nature Communications. I consider this point critical because previous releases by the same group already disclose important synthesis and structural properties of the catalyst, and the current paper fails to demonstrate a genuinely distinct catalytic performance when compared to other alternatives.

I find the work more suitable for publication as a full Research Article in JACS or ACS Catalysis.

Response: Special thanks to the reviewer's nice comment and suggestion. We need to say that the novelty of this work not only focus on tuning the selectivity of catalytic nitriles hydrogenation reaction by structure regulation in atomically dispersed Pd catalysts, but also because of the reduction of nitriles, which is a very delicate cascade reaction system with complex possible products distribution. (Nat Commun, 2021, 12, 3382, <https://doi.org/10.1038/s41467-021-23705-9>; Nat Commun, 2019, 10, 4998, <https://doi.org/10.1038/s41467-019-12993-x>). Therefore, the investigation about the hydrogenation of nitrile is rising as it is a wonderful research mode to understand the selectivity of complex reactions, while atomically dispersed metal catalysts are the bridges that connect the homogeneous catalysts and heterogeneous catalysts.

We acknowledge that we have some previous release that reported the similar nanodiamond-graphene hybrid support (ND@G) based catalyst, but the complexity of nitriles hydrogenation overwhelms other reactions in our previous works, and this work is the first report for tuning the selectivity by atomically dispersed Pd catalysts. We also acknowledge that the reviewer's comment, the fully exposed Pd cluster has somehow similar behaviors with commercial Pd/C, but we have to say that increasing dispersion not just enhance the catalytic efficiency, one reason can be found in our research results that further increasing dispersion to single atom level will alternate the pathway of the reaction. In fact, commercial catalyst, even Raney Ni, has been used for hydrogenating nitriles for a long history. However, from the scientific view, it is far away from exactly understanding the catalytic process on the molecular level due to the structural complexity of the commercial catalysts. Nowadays, it is of great interests that using simple-structural catalyst to delicately understand the catalytic process. Therefore, we think the importance of this work is attractive for the readers of Nature Communications community.

In addition, we fabricated two types of atomically dispersed Pd catalysts on the ND@G support: single Pd atoms ($\text{Pd}_1/\text{ND@G}$) and fully exposed Pd clusters with few Pd atoms ($\text{Pd}_n/\text{ND@G}$). Excitingly, secondary amines and primary amines were obtained with high selectivity by the $\text{Pd}_1/\text{ND@G}$ and $\text{Pd}_n/\text{ND@G}$ catalysts,

respectively, indicating that the reaction paths towards the side products are successfully blocked over these catalysts with atomic precision in structure. Density Functional Theory (DFT) was further employed to establish the relationship of structure sensitivity about hydrogenation of nitriles over these atomically dispersed Pd catalysts. These results indicate that the resident time of the BI surface intermediate is extended on Pd₁-Gr since the difficult activation of the second H₂, which helps the condensation reaction of BI surface intermediate by BA to form the N-benzylidenebenzylamine (DBI) intermediate (DBI eventually generates DBA). On the other hand, the theoretical studies of BN hydrogenation on Pd₃-Gr reveal that the high selectivity of BA originates from the facile activation of the H₂ molecules and the BA weak adsorption after the formation of the BI intermediate.

Indeed, we have disclosed important synthesis and structural properties of the catalyst in our previous work. However, the difference in catalytic behaviors between single Pd atoms and fully exposed Pd clusters in the selectivity of catalytic nitriles hydrogenation reaction has not been studied before our current manuscript. In our manuscript, the structure-performance relationship established over atomically dispersed Pd catalysts provides valuable insights for designing catalysts with tunable selectivity in catalytic hydrogenation of nitriles. We believe that the structure-performance relationship established over the atomically dispersed Pd catalysts in the transfer hydrogenation of nitriles reaction will be of great interest to a broad audience in Nature Communications community.

Special thanks to the reviewer for his/her insightful comments, which really help us a lot in improving the manuscript.

Reviewer #3 (Remarks to the Author):

Comments:

1. Details of how ND@G was prepared is included in the revised manuscript but still I do not see line profiles of ND@G. I understood that the authors studied ND before using XRD, Raman, but I do not understand why these authors cannot show the line profiles for lattice fringes and local diffraction pattern in this study?

Response: Special thanks to the reviewer's nice comment and suggestion. We provided the detailed structure information of the as-prepared ND@G support with nanodiamond core and defective graphene shell as displayed in Figure R1 and R2. As shown in Figure R1a and R1b, the as-prepared ND@G support consists of a well-defined crystalline diamond core with (111) planes (as marked in Figure R1c and R1d) covered by a distorted graphene layer (as marked in Figure R1e and R1f). We are sorry that we provided readers a little bit of information about the ND@G support. We have added our relevant citations (J. Am. Chem. Soc. 2018, 140, 13142, <https://pubs.acs.org/doi/abs/10.1021/jacs.8b07476>; Nat. Commun. 2019, 10, 4431, <https://www.nature.com/articles/s41467-019-12460-7>; ACS Catal. 2019, 9, 5998, <https://pubs.acs.org/doi/abs/10.1021/acscatal.9b00601>) to help readers better understand the structure of ND@G support and re-designed Figure S1 as the reviewer suggested.

Figure R1. (a) HRTEM image of the as-prepared ND@G support with nanodiamond core and defective graphene shell; (b) Structure diagram of ND@G; (c) Partial enlargement HRTEM image focusing on the nanodiamond core and (d) The corresponding line profile of the nanodiamond core; (e) Partial enlargement HRTEM image focusing on the defective graphene shell and (f) The corresponding line profile of the defective graphene shell.

Figure R2. (a) HRTEM image of the as-prepared ND@G support with nanodiamond core and defective graphene shell; (b) Partial enlargement HRTEM image focusing on the defective graphene shell and (c) The corresponding line profile of the defective graphene shell; (d) Partial enlargement HRTEM image focusing on the nanodiamond core and (e) The corresponding line profile of the nanodiamond core.

2. Author explained that the 1.7 nm grain is formed due to the electron exposure in the comment this reviewer. If so, the grains circled by red shown in Fig. 1d are not the true structure of Pd_n/ND@G but it is the one transformed after the exposure of electron. I understood the image shown in Fig. 1d is not the pristine Pd clusters through the authors' reply but without such explanation, it is not accurate and confusing. This arises another question. Why single Pd atoms were able to image as single (Fig. 1b)? If the agglomeration occurs for Pd₃/ND@G, there would be no reason that same agglomeration occurs for Pd₁/ND@G.

Response: We appreciate the reviewer for the nice comment. We are very sorry for not explaining this question clearly. In fact, the catalyst structure observed is not affected by electron exposure, only because of the different projections under the transmission electron microscope, resulting in different morphology, and the clusters in Figure. 1d may exhibit the irregular atomic structures. As we explained in our previous response, the regions that seem to be particles in Figure 1d were island-like aggregates (~1.7 nm) composed of multiple small clusters. In our previous work (ACS Catal. 2019, 9, 5998, <https://pubs.acs.org/doi/abs/10.1021/acscatal.9b00601>; Nat. Commun. 2021,12, 2664, <https://www.nature.com/articles/s41467-021-22948-w>), as shown in Figure R3, the atomic model (a) showing a Pt₃ on the ND@G surface and its projection view (b) along the electron beam direction. The simulated STEM image according to the atomic model

was shown in Figure R3c, where carbon atoms are invisible due to the low contrast. Clearly, the optimized structure of the Pt_3 cluster was not in parallel with carbon support surface, which may explain the irregular atomic structures of Pt clusters under STEM due to the projection nature of transmission electron microscopy technique. In the same way, because the surface of the ND@G support is curved and irregular, Pd species are in different positions on the surface of the carrier, and projection nature of transmission electron microscopy technique, which leads us to see island-like aggregates of large clusters contain smaller clusters at different locations.

To further illustrate the structure of Pd species in $\text{Pd}_n/\text{ND@G}$, AC-HAADF-STEM image of $\text{Pd}_n/\text{ND@G}$ was displayed in Figure R4a, magnified images of the clusters highlighted by yellow circles were shown in Figure R4b and R4c, which is consistent with the simulated STEM images in Figure R2.

We are sorry for not explaining this issue clearly. According to our explanations above, it shows that the structures of single Pd atoms and fully exposed Pd clusters will not be affected by electron exposure, and they have not aggregated. Therefore, the structure we observed in Figure 1 is not inaccurate or confusing.

Figure R3. HAADF-STEM image simulations. Atomic model (a) showing a Pt_3 on defective graphene surface and its projection view (b) along the electron beam direction. (c) Simulated image of the structure in (b) using experimental parameters, where carbon atoms are invisible due to the low contrast (ACS Catal. 2019, 9, 5998, <https://pubs.acs.org/doi/abs/10.1021/acscatal.9b00601>).

Figure R4. AC-HAADF-STEM image (a) of $\text{Pd}_3/\text{ND@G}$ and the magnified images (b, c) of the clusters highlighted by yellow circles in a.

3. Authors mentioned “Although the electron microscopy information is not very clear,

we mainly rely on the analysis of EXAFS data (Table R2) to establish our catalyst model.”. To me, it is hard to agree with this because TEM would give plenty information regarding local atomic structure as this author also showed an atomic resolution TEM (but no line profiles for SACs and the diffraction for ND, which I still consider important to be included). Together with EXAFS, existence of the single atom can be supported but needs careful evaluations. Appl. Phys. Lett. 116, 191903 (2020); <https://doi.org/10.1063/5.0008748>. Notably, EXAFS is not sensitive to highly dispersed metal oxide species with particle sizes <1 nm. Nature Catalysis volume 4, pages453–456 (2021) Relying on the analysis of EXAFS alone would mislead the reader and this is the current problem of the research community. Besides, the suggested k3-weighted EXAFS to show the quality of the measurement is not shown yet, and parameters assumed for example.

Response: We appreciate the reviewer for very nice comment. We really apologize for the inaccurate description in the last response letter. What we want to express is that the AC-HAADF-STEM images at low magnification cannot give detailed structural information. We agree that the TEM can provide plenty information regarding local atomic structure, which is consistent with the reviewer's statement. To further illustrate the structure of Pd species in Pd_n/ND@G, we zoomed in on the particles in Figure R4a, the particles in Pd_n/ND@G catalyst are island-like aggregates composed of multiple small clusters, and there are three Pd atoms in each small cluster. In fact, the DFT model was obtained by combining AC-HAADF-STEM images and EXAFS results. For Pd₁/ND@G (Figure 1a and 1b), all Pd single atoms were uniformly distributed on the support. While in Pd_n/ND@G (Figure 1c and 1d), the Pd species mainly exists as fully exposed Pd clusters. For the electron microscopy morphology of Pd_n/ND@G, we have also made a detailed explanation. Clearly, the Pd species in these two catalysts are both atomically dispersed on the ND@G support. X-ray absorption near-edge structure (XANES) and extended X-ray absorption fine structure (EXAFS) measurements were performed to study the electronic structure and coordination environment of Pd species in Pd₁/ND@G and Pd_n/ND@G catalysts. The EXAFS spectra (Figure 1f) of Pd₁/ND@G only featured a major peak near 1.5 Å from the first coordination shell of Pd associated with Pd-C/O scattering, indicating the formation of isolated Pd atom in Pd₁/ND@G. For Pd_n/ND@G catalyst, the Pd-C/O coordination number was 2.5, while the Pd-Pd coordination number was 1.9, suggesting that the fully exposed Pd cluster was composed by about three Pd atoms on average. The results of XAFS were in good agreement with those of AC-HAADF-STEM, and the structure of the two catalysts was clearly identified.

According to the comments, we carefully reviewed the literatures provided by the reviewer. Indeed, in some cases, EXAFS is not sensitive enough to recognize individual atoms, which may confuse contributions from small clusters or NPs. In addition, the area measured by HAADF-STEM is limited, and other components may be missed (Appl. Phys. Lett. 2020, 116, 191903). Despite the limitations of these characterization techniques, the combination of HAADF-STEM and EXAFS is still the main means of identifying single atom materials. In our work, we carefully studied the results of

HAADF-STEM and EXAFS and combined them to arrive at the structure of the catalyst. At the same time, the XRD results also show that there are no large Pd particles in Pd₁/ND@G and Pd_n/ND@G. Otherwise, in the second literature (Nat. Catal. 2021, 4, 453), we found that the sentence "Notably, EXAFS is not sensitive to highly dispersed metal oxide species with particle sizes <1 nm". However, in our manuscript, the highly dispersed Pd species are in a partial oxidation state rather than a metal oxide, and according to our previous work (J. Am. Chem. Soc. 2018, 140, 13142, <https://pubs.acs.org/doi/abs/10.1021/jacs.8b07476>), the oxygen content on our ND@G support is as low as 3.9 %, which also excludes the possibility of metal oxide formation. Therefore, the EXAFS results in our work are reasonable. More importantly, our model does not rely solely on the results of EXAFS, but carefully evaluates and combines the results of HAADF-STEM and EXAFS, so our results will not mislead readers.

Thank for the good suggestion, we have added the Pd K-edge EXAFS fitting results of Pd₁/ND@G and Pd_n/ND@G in the revised supporting information (please see Figure S5).

4. I appreciate the author to prepare the high mass loading Pd (5wt% one) and performed catalysis in the response to this reviewer, but again it is not characterized well, by for example, high resolution TEM, XRD.

Response: We thank the reviewer for the nice comment. We chose the commercial Pd/C (the Pd weight loading was 5 wt%) as a controlled catalyst, because the Pd nanoparticles in Pd/C catalyst have good metal crystallinity, which can be compared with our two atomically dispersed catalysts to obtain a better verification effect. Figure R5 displays the AC-HAADF-STEM image of commercial Pd/C, the average size of Pd nanoparticles is about 3 nm. We have added the results in the supporting information (Figure S6) .

Figure R5. AC-HAADF-STEM image of the commercial Pd/C catalyst with 5 wt%.

5. TOF is sensitive to the temperature and please provide the information in the vicinity of description of TOF, for example, TOF@300K, otherwise reader needs to look up this information.

Response: Thanks for the good suggestion. We have modified them in the manuscript.

6. I thank the authors effort to take this reviewer's comment account in their work, but I still have strong concerns regarding the quality of material characterization and rationale based on DFT considerations. (there is still large mismatch between observed 1.7 nm Pd NP with Pd₃/ND@G), and have anxious that this work mislead the community. Pd_n/ND@G (blue line) needs to be expressed as Pd₃/ND@G (blue line) since the authors calculate for the latter specific model.

Response: Thanks for the good suggestion. Pd_n/ND@G was used as abbreviation of the cluster catalyst, mainly to distinguish it from the single atoms Pd₁/ND@G catalyst. And we show that the particles in Pd_n/ND@G catalyst are island-like aggregates (~1.7 nm) composed of multiple small clusters, and there are three Pd atoms in each small cluster. Therefore, we use Pd₃-Gr to represent the fully exposed Pd cluster catalyst in DFT calculations. However, to be consistent with the context, we used Pd_n /ND@G as the representative of the cluster catalyst.

7. I recommend the authors to provide more concrete data for the materials used (Pd₁, Pd₃, and Pd_n made by high concentration precursor.).

Response: We thank the reviewer for the comments. In fact, we fabricated two types of atomically dispersed Pd catalysts on the nanodiamond-graphene (ND@G) hybrid support: single Pd atoms (Pd₁/ND@G) and fully exposed Pd clusters with few Pd atoms (Pd_n/ND@G). In order to emphasize the structure of our model in theoretical calculations, we named Pd_n/ND@G as Pd₃-Graphene, so they refer to the same catalyst. We have explained their correspondence and provided comprehensive information in the manuscript and supporting information.

8. In the current version, while this reviewer finds great interest especially the origin of observed selectivity, the provided data is not convincing in my opinion. The other concern is catalysis test condition, which are different as pointed out by the other reviewer. I believe when TOF or conversion efficiency are compared, the system needs to be kept the same for fair comparison and thus, unfortunately the current work needs significant improvements, which impressed me the current work is premature. I believe that it is necessary for the structure to be well analyzed further and its structure and performance to be compared on a one-to-one basis under the same catalysis conditions.

Response: We thank the reviewer for the comments. In the manuscript and supporting information, we provide sufficient and convincing data, including the performance comparison of different catalysts and support, kinetics of two atomically dispersed catalysts, TOF of two atomically dispersed catalysts, hydrogenation properties of different substrates, hydrogenation performance of the two catalysts under different reaction conditions, and comparison of performance of two kinds of atomically

dispersed catalysts and other catalytic systems. And we added the performance evaluation of the catalyst after the reaction to further study the performance of two kinds of atomically dispersed catalysts (Figure S7 in the revised supporting information). The performance of two kinds of atomically dispersed catalysts was studied, and the selectivity regulation strategy established over these catalysts with atomic precision in structure will pave the way for the rational design and construction of the highly selective catalyst with fully metal utilization efficiency.

In fact, different catalysts may have different suitable reaction conditions. Referring to the previous literature (Angew. Chem. Int. Ed. 2016, 55, 14653, <https://doi.org/10.1002/ange.201608345>), we choose the optimal reaction conditions by tuning the reactant concentration and reaction temperature of the two catalysts. In this work, we did not deliberately compare the conversion efficiency of the two catalysts, but focus on the distinct selectivity difference in the catalytic transfer hydrogenation of nitriles, and establish the structure-performance relationship over atomically dispersed Pd catalysts to provide valuable insights for designing catalysts with tunable selectivity.

Special thanks to the reviewer for his/her insightful comments, which really help us a lot in improving the manuscript.

REVIEWER COMMENTS

Reviewer #2 (Remarks to the Author):

After reviewing the final version of the paper, and the answers to the Referees, I consider that the paper can be considered for publication in Nature Communications. I understand that the area may rise sufficient interest, as it explores a hot topic in the literature these days (single site and small metal clusters).

I would however strongly encourage the authors to add new references to seminal work by the Gates's group (UC-Davis), the Flytzani Stephanopoulos's group (Tufts Univ), and the Corma's group (ITQ), as well as people working in the NO_x reduction and methane to methanol spaces (Lercher, Gounder, Roman-Lerskov, etc).

A few possibilities (of course, only suggestions) that I consider relevant in the context of the last response provided to this Reviewer:

- J. Am. Chem. Soc. 2011, 133, 13, 4714–4717
- J. Am. Chem. Soc. 2015, 137, 10, 3470–3473
- Nature Communications volume 6, Article number: 7546 (2015)
- Angewandte Chemie, 60, 29, 2021, 15954-15962

Reviewer #3 (Remarks to the Author):

I see some improvements in the revised manuscript but unfortunately, still I have strong concern regarding the Pdn cluster. In the revised manuscript, Pd₃ was changed to Pdn but it does not make sense. it is not FECC, too big to compare with Pd₁ (see my previous comment #6) and DFT is still for the Pt₃ cluster which does not represent this study(see previous comment #10). I respect the authors effort for the improvements of their work, but the manuscript is not revised appropriately, and seems hard to convince this reviewer since the conflicts this reviewer pointed out previously is not sufficiently solved. It needs more systematic and fair comparisons with well-defined size selected smaller size Pdn nanoclusters

Reviewer #2 (Remarks to the Author):

After reviewing the final version of the paper, and the answers to the Referees, I consider that the paper can be considered for publication in Nature Communications. I understand that the area may rise sufficient interest, as it explores a hot topic in the literature these days (single site and small metal clusters). I would however strongly encourage the authors to add new references to seminal work by the Gates's group (UC-Davis), the Flytzani Stephanopoulos's group (Tufts Univ), and the Corma's group (ITQ), as well as people working in the NO_x reduction and methane to methanol spaces (Lercher, Gounder, Roman-Lershkov, etc). A few possibilities (of course, only suggestions) that I consider relevant in the context of the last response provided to this Reviewer:

J. Am. Chem. Soc. 2011, 133, 13, 4714–4717; J. Am. Chem. Soc. 2015, 137, 10, 3470–3473; Nature Communications volume 6, Article number: 7546 (2015); Angewandte Chemie, 60, 29, 2021, 15954-15962

Response: We thank the reviewer for the nice suggestion. We referenced relevant significant studies by Gates's group, Flytzani Stephanopoulos's group, Corma's group and Lercher's group. (Please see references : 22, 28, 29, 30)

Special thanks to the reviewer for his/her insightful comments, which really help us a lot in improving the manuscript.

Reviewer #3 (Remarks to the Author):

I see some improvements in the revised manuscript but unfortunately, still I have strong concern regarding the Pd_n cluster. In the revised manuscript, Pd₃ was changed to Pd_n but it does not make sense. it is not FECC, too big to compare with Pd₁ (see my previous comment #6) and DFT is still for the Pd₃ cluster which does not represent this study (see previous comment #10). I respect the authors effort for the improvements of their work, but the manuscript is not revised appropriately, and seems hard to convince this reviewer since the conflicts this reviewer pointed out previously is not sufficiently solved. It needs more systematic and fair comparisons with well-defined size selected smaller size Pd_n nanoclusters

Response: We thanks very much for the reviewer's nice comment. We fell really sorry for still not explaining this question clearly to the reviewer. The reviewer concerned "the diameter of Pd_n nanocluster in Figure. 1 d is about 1.7 nm and it should consist of tens of Pd atoms, but not Pd₃ cluster".

In order for the reviewer to better understand this Pd_n structure, the AC-HAADF-STEM images of two clusters marked as A and B in Figure R1d were zoomed up and displayed in Figure R2. The size of these two clusters indeed seems to be about 1.7 nm judged from Figure R1d, but it can be noted from Figure R2 that this "1.7 nm" atom island is actually the random aggregation of several small Pd cluster unit highlighted in yellow circles in Figure R2.

Because the projection nature of transmission electron microscopy technique as shown in Figure R3 and these small Pd clusters marked in Figure R2 are randomly located on the curved surface of the ND@G, which make them present island-like aggregates of large clusters around 1.7 nm, actually they are composed by several smaller Pd clusters with average three Pd atoms (Pd₃ clusters), which was further identified by XAFS analysis (Figure R1e and R1f, Table R1). The other strong evidence is that the Pd-Pd coordination number was only 1.9 in Pd_n cluster,

suggesting that the smaller Pd cluster unit highlighted in yellow circles (Figure R2) was composed by about three Pd atoms on average.

Thus, according to the AC-STEM and EXAFS analysis, the three-atom Pd cluster on graphene layer ($\text{Pd}_3\text{-G}$) was constructed to represent the active sites on $\text{Pd}_n/\text{ND}@G$ for the DFT calculation. Actually, the similar structure model of Pt was also captured and identified in recent studies (ACS Catal. 2020, 10, 12696, <https://pubs.acs.org/doi/full/10.1021/acscatal.0c03464>; Nat. Commun. 2021, 12, 2664, <https://www.nature.com/articles/s41467-021-22948-w>; Small 2021, 17, 2100732, <https://doi.org/10.1002/sml.202100732>).

Figure. R1 (a) HAADF-STEM images of $\text{Pd}_1/\text{ND}@G$ at low magnification. (b) Atomically dispersed single Pd atoms in $\text{Pd}_1/\text{ND}@G$ highlighted by the yellow circles. (c) HAADF-STEM images of $\text{Pd}_n/\text{ND}@G$ at low magnification. (d) Fully exposed Pd clusters in $\text{Pd}_n/\text{ND}@G$ highlighted by the red circles. (e) Pd K-edge XANES profiles and (f) EXAFS spectra for $\text{Pd}_1/\text{ND}@G$ and $\text{Pd}_n/\text{ND}@G$.

Figure R2. The magnified AC-HAADF-STEM images of the cluster A and cluster B as marked in Figure R1, the Pd_3 cluster was highlighted by yellow circles.

Figure R3. HAADF-STEM image simulations. Atomic model (a) showing a Pt₃ on defective graphene surface and its projection view (b) along the electron beam direction. (c) Simulated image of the structure in (b) using experimental parameters, where carbon atoms are invisible due to the low contrast (ACS Catal. 2019, 9, 5998, <https://pubs.acs.org/doi/abs/10.1021/acscatal.9b00601>).

Table R1. Structural parameters extracted from quantitative EXAFS curve-fitting ($S_0^2 = 0.8 - 0.9$).

Sample	Shell	C.N. ^[a]	R(Å) ^[b]	$\Delta\sigma^2/(10^{-3} \text{ \AA}^2)$ ^[c]	ΔE_0 (eV) ^[d]	R factor
Pd foil	Pd-Pd	12.0	2.74	5.86	-6.49	0.001
PdO	Pd-O	4.0	2.01	2.13	-2.62	0.001
Pd ₁ /ND@G	Pd-Pd	0	-	-	-	0.027
	Pd-C/O	2.6	2.04	1.80	4.660	
Pd _n /ND@G	Pd-Pd	1.9	2.75	5.76	8.603	0.030
	Pd-C/O	2.5	2.02	5.19		

[a] C.N. is the coordination number. [b] R is interatomic distance (the bond length between Pd central atoms and surrounding coordination atoms). [c] σ^2 is Debye-Waller factor (a measure of thermal and static disorder in absorber scatterer distances).[d] ΔE_0 is edge energy shift (the difference between the zero kinetic energy value of the sample and that of the theoretical model).

Special thanks to the reviewer for his/her insightful comments, which really help us a lot in improving the manuscript.

REVIEWERS' COMMENTS

Reviewer #3 (Remarks to the Author):

I thank the authors efforts again in the response to my comments. Figure R2 is convincing and now my concerns seems fully solved. On publication, I strongly recommend including Fig. R2 in manuscript since this is the motivation why this author performed DFT with Pd₃ and following explanation written in the previous authors' response letter.

“Because the projection nature of transmission electron microscopy technique as shown in Figure R3 and these small Pd clusters marked in Figure R2 are randomly located on the curved surface of the ND@G, which make them present island-like aggregates of large clusters around 1.7 nm, actually they are composed by several smaller Pd clusters with average three Pd atoms (Pd₃ clusters), which was further identified by XAFS analysis (Figure R1e and R1f, Table R1). The other strong evidence is that the Pd-Pd coordination number was only 1.9 in Pd_n cluster, suggesting that the smaller Pd cluster unit highlighted in yellow circles (Figure R2) was composed by about three Pd atoms on average.

Thus, according to the AC-STEM and EXAFS analysis, the three-atom Pd cluster on graphene layer (Pd₃-G) was constructed to represent the active sites on Pd_n/ND@G for the DFT calculation. Actually, the similar structure model of Pt was also captured and identified in recent studies (ACS Catal. 2020, 10, 12696, <https://pubs.acs.org/doi/full/10.1021/acscatal.0c03464>; Nat. Commun. 2021, 12, 2664, <https://www.nature.com/articles/s41467-021-22948-w>; Small 2021, 17, 2100732, <https://doi.org/10.1002/sml.202100732>).

”

I believe all information above is required to justify the approach the authors took and convince the readership of this journal. Without this, it will be unclear if it is Pd₃ and why authors performed DFT with Pd₃.

Reviewer #3 (Remarks to the Author):

I thank the authors efforts again in the response to my comments. Figure R2 is convincing and now my concerns seem fully solved. On publication, I strongly recommend including Fig. R2 in manuscript since this is the motivation why this author performed DFT with Pd₃ and following explanation written in the previous authors' response letter.

“Because the projection nature of transmission electron microscopy technique as shown in Figure R3 and these small Pd clusters marked in Figure R2 are randomly located on the curved surface of the ND@G, which make them present island-like aggregates of large clusters around 1.7 nm, actually they are composed by several smaller Pd clusters with average three Pd atoms (Pd₃ clusters), which was further identified by XAFS analysis (Figure R1e and R1f, Table R1). The other strong evidence is that the Pd-Pd coordination number was only 1.9 in Pd_n cluster, suggesting that the smaller Pd cluster unit highlighted in yellow circles (Figure R2) was composed by about three Pd atoms on average.

Thus, according to the AC-STEM and EXAFS analysis, the three-atom Pd cluster on graphene layer (Pd₃-G) was constructed to represent the active sites on Pd_n/ND@G for the DFT calculation. Actually, the similar structure model of Pt was also captured and identified in recent studies (ACS Catal. 2020, 10, 12696, <https://pubs.acs.org/doi/full/10.1021/acscatal.0c03464>; Nat. Commun. 2021, 12, 2664, <https://www.nature.com/articles/s41467-021-22948-w>; Small 2021, 17, 2100732, <https://doi.org/10.1002/sml.202100732>). ”

I believe all information above is required to justify the approach the authors took and convince the readership of this journal. Without this, it will be unclear if it is Pd₃ and why authors performed DFT with Pd₃.

Response: We thanks very much for the reviewer's nice comment. We added the previous Fig. R2 in the supporting information and the brief explanation was also addressed.

Figure R2. The magnified AC-HAADF-STEM images of the cluster A and cluster B as marked in Figure R1, the Pd₃ cluster was highlighted by yellow circles.